# SDMG: Smoothing Your Diffusion Models for Powerful Graph Representation Learning

Junyou Zhu [1 2]   Langzhou He [3]   Chao Gao [4]   Dongpeng Hou [4]   Zhen Su [1]   Philip S. Yu [3]   Jürgen Kurths [1]
Frank Hellmann [1]

## Abstract

Diffusion probabilistic models (DPMs) have recently demonstrated impressive generative capabilities. There is emerging evidence that their sample reconstruction ability can yield meaningful representations for recognition tasks. In this paper, we demonstrate that the objectives underlying generation and representation learning are not perfectly aligned. Through a spectral analysis, we find that minimizing the mean squared error (MSE) between the original graph and its reconstructed counterpart does not necessarily optimize representations for downstream tasks. Instead, focusing on reconstructing a small subset of features, specifically those capturing global information, proves to be more effective for learning powerful representations. Motivated by these insights, we propose a novel framework, the Smooth Diffusion Model for Graphs (SDMG), which introduces a multi-scale smoothing loss and low-frequency information encoders to promote the recovery of global, low-frequency details, while suppressing irrelevant t high-frequency noise. Extensive experiments validate the effectiveness of our method, suggesting a promising direction for advancing diffusion models in graph representation learning.

[1]Department of Complexity Science, Potsdam Institute for Climate Impact Research, 14473 Potsdam, Germany [2]Machine Learning Group, Technical University of Berlin, 10587 Berlin, Germany [3]Department of Computer Science, University of Illinois at Chicago, Chicago, IL 60607, USA [4]School of Artificial Intelligence, Optics and Electronics (iOPEN), Northwestern Polytechnical University, Xi'an 710072, China. Correspondence to: Chao Gao <cgao@nwpu.edu.cn>.

*Proceedings of the 42nd International Conference on Machine Learning*, Vancouver, Canada. PMLR 267, 2025. Copyright 2025 by the author(s).

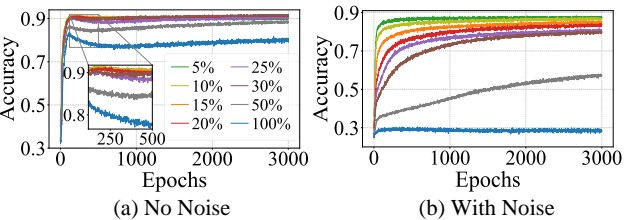

(a) No Noise          (b) With Noise

*Figure 1.* Impact of reconstructing various proportions of low-frequency components on node classification for *Photo*. (a) We sort the frequency components (derived from the graph Laplacian) in ascending order and reconstruct only the lowest fraction (e.g., 5%) of the frequency components. This partial reconstruction already achieves high accuracy, while restoring the full spectrum (100%) overfits high-frequency noise. (b) Under Gaussian noise, focusing on a narrow low-frequency band remains robust, whereas full-spectrum reconstruction collapses in performance.

## 1. Introduction

Self-supervised learning (SSL) is transforming representation learning by reducing our reliance on massive labeled datasets (Yu et al., 2024b; Xiao et al., 2024; Kalantidis et al., 2020; Zhu et al., 2022). Among its many variants, generative diffusion probabilistic models (DPMs) (Ho et al., 2020; Song et al., 2021) have emerged as particularly powerful: by reconstructing clean signals from corrupted inputs, these models capture rich semantic insights. Recent vision research shows that stronger diffusion models yield more expressive embeddings (Hudson et al., 2024; Wang et al., 2023; Wei et al., 2023). This has also spurred adoption in more general graph domains. A pioneering example, the DDM (Yang et al., 2024), deploys directional perturbations to denoise node features, revealing how diffusion-based techniques can elegantly handle complex, anisotropic graph structures.

Despite the growing popularity of diffusion-based graph representation methods, existing approaches (Yang et al., 2024) adopt off-the-shelf DPMs tailored for *generation* (Niu et al., 2020), leaving open the question of whether the generation goal to fully reconstruct every detail of a graph truly enhances the discriminative power of representations. However, our empirical findings (Figure 1(a)) suggest: focusing

on reconstructing only a narrow band of low-frequency information (e.g., 5%) can already yield strong embeddings, while attempting to fully restore all features (i.e., 100%) often proves unnecessary—or, even worse, detrimental. We push this point further in Figure 1(b) by adding task-irrelevant Gaussian noise to node features; finding that reconstructing only low-frequency components preserves a competitive accuracy even if the data is considerably perturbed, whereas insisting on reconstructing everything causes a performance to drop significantly. These observations highlight a misalignment between generation-oriented objectives and representation learning. This raises a fundamental tension: should we aim to faithfully recover all graph details, or strategically suppress unnecessary details to better emphasize semantically meaningful signals for downstream tasks?

To investigate this, we begin with an empirical study (Section 3) by analyzing how different frequency components affect classification performance. Our analysis reveals two main insights: **(1)** aggressively minimizing the MSE-based generation-reconstruction loss, particularly for high-frequency details, can undermine downstream accuracy, and **(2)** focusing the model on a narrow range of low-frequency signals results in stronger representations and faster convergence. In fact, reconstructing additional high-frequency structures introduces distracting noise for downstream tasks, even though it is necessary and beneficial for generation. This aligns with established GNN research emphasizing the predictive power of low-frequency features (Hoang et al., 2021; Liu et al., 2022). Although one might consider incorporating GNNs as low-pass filters into the denoising encoder, we find that high-frequency components are inevitably re-learned and encoded as we continue to minimize the MSE-based reconstruction objective (demonstrated in Figure 4 of Section 3). This issue remains challenging, especially since standard constraints designed for filtering noise frequency components, like information bottlenecks (Yu et al., 2024a), rely on labeled supervision, which is not readily available in our SSL setting.

Motivated by these findings, we propose **S**mooth **D**iffusion **M**odel for **G**raphs (SDMG), a novel self-supervised framework designed to learn recognition-oriented representations without labels. SDMG employs two dedicated low-frequency encoders, one for node features and another for topology, to distill global, low-frequency signals. To avoid reintroducing unhelpful high-frequency details, we propose a new multi-scale smoothing objective. Rather than strictly enforcing a pointwise reconstruction, our objective aligns the original and reconstructed graphs at multiple scales, theoretically encouraging the model to emphasize low-frequency signals while suppressing high-frequency noise. This strategy effectively "filters out" distracting details, allowing the diffusion-based representation learner to

focus on semantically meaningful structures aligned with downstream recognition tasks. Our key contributions include:

- We theoretically and empirically uncover that purely generation-oriented objectives can conflict with recognition goals, revealing how excessive high-frequency reconstruction reduces the representation quality.

- We propose the Smooth Diffusion Model for Graphs (SDMG), an approach that explicitly leverages low-frequency filters and a novel *multi-scale smoothing* (MSS) loss to align pre-training reconstruction with downstream recognition. To the best of our knowledge, we are the first to address this misalignment between graph generation and representation.

- Extensive node- and graph-level experiments show that SDMG achieves state-of-the-art performance, pointing to a promising direction for diffusion-based graph SSL.

## 2. Preliminaries

### 2.1. Notations

Consider a graph $G = (\mathcal{V}, \mathcal{E})$ with $|\mathcal{V}| = N$ and the adjacency matrix $\mathbf{A} \in \mathbb{R}^{N \times N}$, where $A_{ij} \in \{0, 1\}$ indicates the presence of an edge between nodes $i$ and $j$, and $N$ denotes the number of nodes. Let $\mathbf{D} = diag(deg_1, \ldots deg_N)$ denote the degree matrix, where $deg_i = \sum_{j \in \mathcal{V}} A_{ij}$. For convenience, we define the augmented adjacency matrix $\tilde{\mathbf{A}} = \mathbf{A} + \mathbf{I}$ and its degree matrix $\tilde{\mathbf{D}} = \mathbf{D} + \mathbf{I}$. With these, we construct the normalized adjacency matrix $\mathbf{A}_{\text{norm}} = \tilde{\mathbf{D}}^{-1/2} \tilde{\mathbf{A}} \tilde{\mathbf{D}}^{-1/2}$. The symmetric normalized Laplacian matrix is $\hat{\mathbf{L}} = \mathbf{I} - \mathbf{A}_{\text{norm}}$. Since $\hat{\mathbf{L}}$ is symmetric normalized, its eigen-decomposition is $\mathbf{U}\mathbf{\Lambda}\mathbf{U}^{\top}$, where $\mathbf{\Lambda} = diag(\hat{\lambda}_1, \ldots, \hat{\lambda}_N)$ and $\mathbf{U} = [\mathbf{u_1}^{\top}, \ldots, \mathbf{u_N}^{\top}] \in \mathbb{R}^{N \times N}$ are the eigenvalues and eigenvectors of $\hat{\mathbf{L}}$, respectively. Each node $v_i \in \mathcal{V}$ is associated with a feature vector $x_i \in \mathbb{R}^d$, and we stack all node features into $\mathbf{X} = [x_1, x_2, \ldots, x_N] \in \mathbb{R}^{N \times d}$. Then, the graph Fourier transform based on $\hat{\mathbf{L}}$ is represented by $\tilde{\mathbf{X}} = \mathbf{U}^{\top} \tilde{\mathbf{D}}^{1/2} \mathbf{X}$ and the inverse transform is $\mathbf{X} = \tilde{\mathbf{D}}^{-1/2} \mathbf{U} \tilde{\mathbf{X}}$.

### 2.2. Diffusion Models

DPMs define a forward-noising process and a corresponding reverse-time denoising process to generate samples from complex distributions (Ho et al., 2020; Song et al., 2021; Cho et al., 2024). Formally, consider a continuous-time stochastic process $\{\mathbf{x}_t\}_{t=0}^{T}$ on $\mathbb{R}^d$. The forward process gradually perturbs an initial data point $\mathbf{x}_0$ by noise via:

$$\mathrm{d}\mathbf{x}_t = f(t)\mathbf{x}_t \, \mathrm{d}t + g(t) \, \mathrm{d}\mathbf{w}_t, \tag{1}$$

where $\mathbf{w}_t$ is a standard Wiener process, and $f(t), g(t)$ dictate the drift and diffusion terms, respectively. As $t \to T$,

$\mathbf{x}_T$ converges to a Gaussian distribution independent of the initial input.

By reversing time, one obtains a reverse SDE:

$$\mathrm{d}\mathbf{x}_t = [f(t)\mathbf{x}_t - g^2(t)\nabla_\mathbf{x} \log p_t(\mathbf{x}_t)]\,\mathrm{d}t + g(t)\,\mathrm{d}\bar{\mathbf{w}}_t, \quad (2)$$

where $\bar{\mathbf{w}}_t$ is a reverse-time Wiener process, and $\nabla_\mathbf{x} \log p_t(\mathbf{x}_t)$ is the score function. Approximating this score enables simulating the reverse dynamics to recover clean samples from noise.

In practice, a neural network $\mathbf{x}_\psi(\mathbf{x}_t, t)$ with parameters $\psi$ approximates the score or directly predicts $\mathbf{x}_0$. A common objective is a weighted mean-squared error between $\mathbf{x}_\psi(\mathbf{x}_t, t)$ and $\mathbf{x}_0$, known as the denoising score matching (DSM) loss:

$$\mathcal{L}_{\mathrm{DSM}} = \mathbb{E}_t \left\{ \lambda(t)\, \mathbb{E}_{\mathbf{x}_0} \mathbb{E}_{\mathbf{x}_t|\mathbf{x}_0} \left[ \|\mathbf{x}_\psi(\mathbf{x}_t, t) - \mathbf{x}_0\|^2 \right] \right\}. \quad (3)$$

While DPMs have achieved remarkable success in Euclidean domains, their application to graph representation learning is still emerging. DDM (Yang et al., 2024) is the first work to integrate diffusion into node feature spaces and employ a denoising decoder (a U-Net architecture) to recover $\mathbf{x}_0$ from $\mathbf{x}_t$. The latent representations $\mathbf{H}$ are then extracted from intermediate layers of this decoder:

$$\mathbf{H} = h_\omega(\mathbf{x}_t), \quad (4)$$

where $h_\omega$ denotes the encoder portion of the denoising network at a chosen intermediate layer. However, we argue that strictly generative objectives in DDM may not align with discriminative goals, prompting us to rethink the training strategy for improved representation learning.

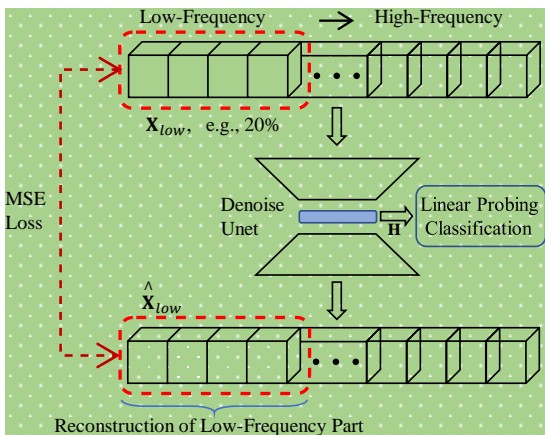

*Figure 2.* A case study model illustrates how only the lowest proportions of the frequency components (green boxes) that are extracted from node features or topology are reconstructed.

# 3. An Investigation into the Effects of Smooth Global Signal Reconstruction

We begin by empirically and theoretically examining how the reconstruction of different frequency components in graph signals impacts the performance on downstream tasks (e.g., node classification). Our study uncovers two findings:

**Finding 1: Reconstructing Low-Frequency Components Suffices for High-Quality Representations.** We investigate which frequency components most benefit representation learning by *selectively reconstructing* only the lower end of the spectrum (see Figure 2). We first express the node-feature matrix $\mathbf{X}$ in the spectral domain based on the graph Fourier transform as

$$\tilde{\mathbf{X}} = \mathbf{U}^\top \tilde{\mathbf{D}}^{1/2}\mathbf{X} \in \mathbb{R}^{N \times d} \quad (5)$$

where $\mathbf{U}$ and $\tilde{\mathbf{D}}$ are defined in Section 2.1. To retain only the $q$ smallest eigenvalues (i.e., the lowest-frequency components), we conduct a low-frequency truncation. Specifically, $\mathbf{U}_{(q)} := [\mathbf{u}_1, \ldots, \mathbf{u}_q] \in \mathbb{R}^{N \times q}$ denote the sub-matrix that collects the first $q$ eigenvectors (i.e. the eigenmodes with the smallest eigenvalues). Projecting onto this subspace yields the retained coefficients

$$\tilde{\mathbf{X}}_{(q)} = \mathbf{U}_{(q)}^\top \tilde{\mathbf{D}}^{1/2}\mathbf{X} \in \mathbb{R}^{q \times d}, \quad (6)$$

while all higher-frequency rows $q + 1{:}N$ are discarded. We then map $\tilde{\mathbf{X}}_{(q)}$ back to the spatial domain via

$$\mathbf{X}_{\mathrm{low}} = \tilde{\mathbf{D}}^{-1/2}\mathbf{U}_{(q)}\tilde{\mathbf{X}}_{(q)}, \quad (7)$$

thus obtaining a feature matrix $\mathbf{X}_{\mathrm{low}}$ that captures only the globally smooth, low-frequency signal. Varying $q$ smoothly controls the amount of high-frequency detail that is removed.

With $\mathbf{X}_{\mathrm{low}}$ in hand, we feed these low-frequency node features into a vanilla diffusion-based denoising model without using any low-frequency encoders (e.g., GNNs) but using MLPs and extracting intermediate-layer representations for linear probing in node classification.

Figure 3 shows the results on two real-world datasets. the accuracy rises sharply when the model reconstructs roughly the lowest 20 % of the entire frequency spectrum (orange region). Once components beyond this lowest-frequency band are added (green region), performance levels off or even declines. In other words, recovering only the bottom fifth of the spectrum already captures most of the discriminative signal, whereas reconstructing additional higher-frequency components brings little benefit and can be detrimental.

**Finding 2: Overemphasizing MSE Minimization Reintroduces Irrelevant Details.** Simply pushing the MSE

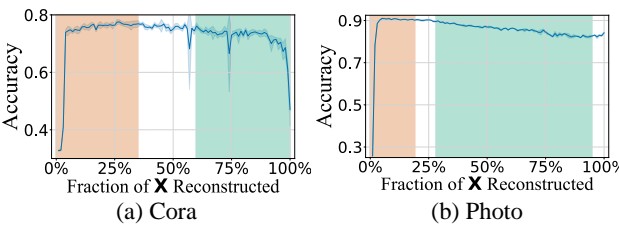

*Figure 3.* Node–classification accuracy when we only rebuild the low-frequency part of the feature matrix $\mathbf{X}$. We keep the first $q$ Laplacian eigenmodes, reconstruct $\mathbf{X}$ from these components, and plot the ratio $q/N$ on the $x$-axis ($N$ = total number of modes, so $q/N = 1$ means full-spectrum reconstruction). Accuracy peaks for small $q$ (orange band) and falls as higher-frequency components are progressively added (green band). "$\mathbf{X}$" denotes node-feature reconstruction.

as low as possible can unintentionally force the model to encode high-frequency or noisy features that do not help, and can even harm, downstream classification. Below, we first provide theoretical evidence and then empirical results to support this point.

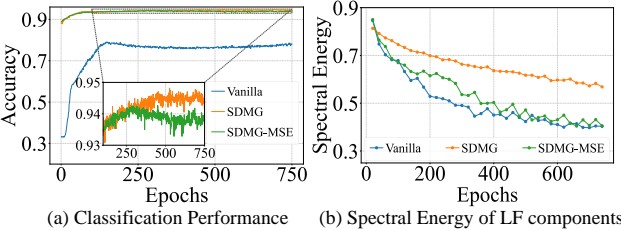

*Figure 4.* Performance and spectral energy for three model variants on *Photo* datasets. (a) Node classification accuracy for *Vanilla* diffusion, *SDMG* (with low-frequency encoders and MSS loss), and *SDMG-MSE* (with low-frequency encoders but MSE loss). (b) The spectral energy of the lowest 40% frequency components, where higher values indicate better low-frequency (LF) capture.

**Theorem 3.1.** *Consider an encoding $\mathbf{Z}$ of $\mathbf{X}$ with a bounded capacity $I(\mathbf{X}; \mathbf{Z}) \leq C$, where $I(\cdot; \cdot)$ denotes mutual information. Assume that $\mathbf{X}$ takes values in the unit ball of an Euclidean space. Let $Y$ be a target variable to be predicted from $\mathbf{X}$, and assume $\mathbf{X}$ can be decomposed as $\mathbf{X} = (\mathbf{X}_s, \mathbf{X}_r)$ such that $\mathbf{X}_r$ is relevant to $Y$ and $\mathbf{X}_s$ is superfluous to predicting $Y$ given $\mathbf{X}_r$, that is $I(\mathbf{X}_s; Y | \mathbf{X}_r) = 0$. Then we have:*

*The minimal mean square error (mmse) for predicting $\mathbf{X}$ from $\mathbf{Z}$ is bounded as*

$$\mathrm{mmse}(\mathbf{X} \mid \mathbf{Z}) \geq \mathrm{var}(\mathbf{X}) - \frac{1}{2}\Big(I(\mathbf{Z}; \mathbf{X}_r) + I(\mathbf{Z}; \mathbf{X}_s \mid \mathbf{X}_r)\Big). \quad (8)$$

*The information that the encoding $\mathbf{Z}$ carries on the variable $Y$ is bounded by*

$$I(\mathbf{Z}; Y) \leq C - I(\mathbf{Z}; \mathbf{X}_s | \mathbf{X}_r) \quad (9)$$

*Therefore, optimizing the encoding $\mathbf{Z}$ in order to minimize $mmse(\mathbf{X}|\mathbf{Z})$ is partially misaligned with optimizing the encoding for $I(\mathbf{Z}; Y)$. The former benefits from an increase in encoded superfluous information, the latter is penalized by it.*

The proof is in Appendix A.1. Theorem 3.1 indicates that as the portion of the encoder's capacity allocated to irrelevant information increases, the model's ability to encode label-relevant information diminishes.

Empirically, Figure 4 compares three diffusion variants on the *Photo* dataset: *(i) Vanilla*, trained purely with MSE; *(ii) SDMG-MSE*, which adds low-frequency encoders but still minimizes MSE; *(iii) SDMG*, our method which employs low-frequency encoders with our multi-scale smoothing loss. Panel (a) reports classification accuracy, while panel (b) tracks low-frequency spectral energy. Three observations stand out:

**Observation 1: Vanilla overfits.** Without a low-frequency filter, the model struggles to capture only low-frequency components, resulting in a substantial performance gap. Moreover, the MSE objective eventually overfits high-frequency details, causing an accuracy drop after around 100 epochs.

**Observation 2: Low-frequency encoders boost early gains.** By introducing low-frequency GNN encoders, *SDMG-MSE* and *SDMG* quickly captures global signals, achieving nearly 88% accuracy by epoch 0, far surpassing the baseline initialization.

**Observation 3: Pure MSE reintroduces noise.** Despite the initial gains, *SDMG-MSE* eventually exhibits the downward trend of *Vanilla* in later epochs. Because MSE penalizes errors in all frequency bands equally, the model eventually reconstructs high-frequency details, causing a drop in classification accuracy. This is further demonstrated by Figure 4(b), which shows that *SDMG-MSE* converges toward the same (reduced) low-frequency energy level as *Vanilla*, indicating an increased emphasis on reconstructing unhelpful high-frequency signals.

### 3.1. Broader Implications and Insights

Although high-frequency details may occasionally help (Figure 3(b)), our results show that low-frequency, global signals are more crucial for graph tasks. Prioritizing these smooth components over complete reconstruction is more effective under limited capacity (Section 4). Future work could explore *frequency-adaptive* objectives to focus on relevant spectral bands for downstream tasks.

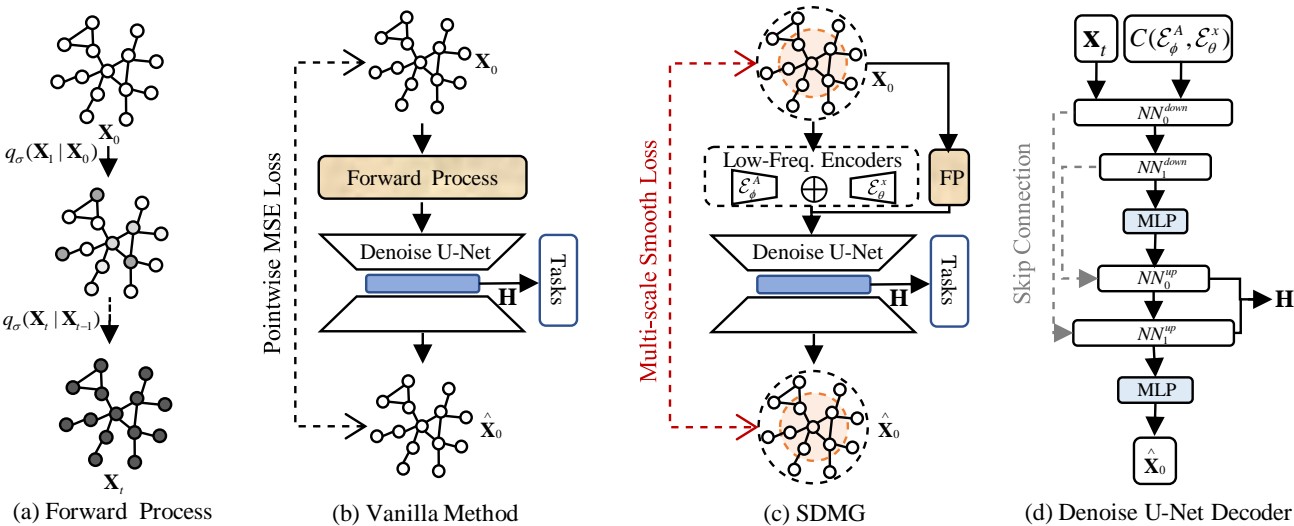

*Figure 5.* Overview of the SDMG framework. (a) Forward diffusion: Gaussian noise is incrementally added to the original sample $\mathbf{X}_0$ to obtain the corrupted version $\mathbf{X}_t$. (b) Standard diffusion model: $\mathbf{X}_t$ is input into a denoise U-Net decoder, which reconstructs using an MSE loss, with activations from specific U-Net layers serving as representations $\mathbf{H}$. (c) SDMG enhancements: Low-frequency encoders ($\mathcal{E}_\phi^A$, and $\mathcal{E}_\theta^x$) extract information from node features and graph topology as conditional inputs for the U-Net (Section 4.2), and a new multi-scale smoothing loss prioritizes the reconstruction of important low-frequency information (Section 4.3). (d) Customized conditional U-Net architecture designed for diffusion-based graph representation learning (Section 4.3 and Appendix D).

# 4. Method: Smooth Diffusion Model for Graph Representation Learning

Current diffusion model-based graph representation learning faces the *challenge* of misalignment between generative and recognition objectives. As shown in Section 3, focusing on low-frequency components of node features and adjacency matrices significantly alleviates this misalignment, thereby improving downstream performance. However, computing the low-frequency embeddings via graph Fourier transforms (Equations (6) and (7)) is still computationally expensive, which makes it impractical for large-scale graphs.

To address this, we propose learnable encoders that efficiently approximate low-frequency embeddings. In **Section 4.2**, we introduce these encoders to extract global low-frequency information, which enhances an early performance. However, despite the presence of these encoders, we observe that a pure MSE-based reconstruction objective eventually reintroduces high-frequency noise (evidences in Figure 4).

To mitigate this, **Section 4.3** proposes a new multi-scale loss that emphasizes low-frequency reconstruction, better aligning generation with recognition tasks. Additionally, **Section 4.1** details the reconstruction objective, and Section 4.3 introduces a denoising decoder. The SDMG framework is illustrated in Figure 5.

## 4.1. Reconstruction Objective

Graph data consists of node features and topology, each providing distinct perspectives for reconstruction. In self-supervised learning, we need to decide which part of this signal to reconstruct. Note that in most real-world graphs the adjacency matrix is extremely sparse. The number of present edges grows linearly with $|\mathcal{V}|$, whereas the number of absent edges grows quadratically. This produces a severe class imbalance if every entry of $\mathbf{A}$ is treated as a regression target. Node features, on the other hand, are typically dense and directly related to downstream prediction tasks. We therefore formulate the denoising objective solely with respect to the clean feature matrix $\mathbf{X}$:

$$\mathcal{L}_{\text{rec}} = \mathbb{E}_{t,\mathbf{x}_0}\left[\lambda(t)\left\|\mathbf{x}_\psi(\mathbf{x}_t, t, C) - \mathbf{x}_0\right\|^2\right], \quad (10)$$

where $\mathbf{x}_t$ is the noised input at time $t$, $\mathbf{x}_\psi$ the denoiser and $C$ a conditioning term described next.

Although we do not regress the full adjacency, structural patterns are indispensable for high-quality representations (HaoChen et al., 2021). Instead of predicting every edge entry, we distil the global, low-frequency information of the graph Laplacian and feed it to the denoiser as context:

$$C = \big(\underbrace{\mathcal{E}_\phi^A(\hat{\mathbf{L}})}_{\text{topology LF}}, \underbrace{\mathcal{E}_\theta^X(\mathbf{X})}_{\text{feature LF}}\big),$$

where $\mathcal{E}_\phi^A$ and $\mathcal{E}_\theta^X$ are encoders for graph topology and node features, respectively, which will be detailed in Seciton 4.2.

This design lets the model exploit long-range structural regularities without being overwhelmed by the highly imbalanced edge-level target. By defining $C$ as a concatenation function, our model incorporates the low-frequency components of both node features and adjacency matrices as conditioning information during reconstruction, as described in Section 4.2.

## 4.2. Low-Frequency Component Encoders

**Graph Topology Low-frequency Information Encoder** $\mathcal{E}_\phi^A$**.** We aim to approximate the first $q$ eigenvectors of the normalized Laplacian $\hat{\mathbf{L}}$ using a neural network function $\mathcal{E}_\phi^A$. Specifically, the encoder $\mathcal{E}_\phi^A$, implemented as a multi-layer perceptron, takes node-level structural features (e.g. random-walk positional encodings derived from the adjacency matrix (Rampášek et al., 2022)) and outputs an embedding matrix $\mathcal{A} \in \mathbb{R}^{N \times q}$. The key insight is that by minimizing an appropriate loss, we can recover the low-frequency eigenvectors $\mathbf{U}_{(q)}$. The optimization objective for the encoder is then:

$$\min_{\mathcal{A} \in \mathbb{R}^{N \times q}} \mathcal{L}(\mathcal{A}) := \left\| \hat{L} - \mathcal{A}\mathcal{A}^\top \right\|_F^2, \tag{11}$$

Based on the classical low-rank approximation results (Eckart–Young–Mirsky theorem (Eckart & Young, 1936)), this objective ensures that $\mathcal{A}$ approximates the smallest $q$ eigenvectors of the graph Laplacian up to a scaling factor.

**Node Feature Low-frequency Information Encoder** $\mathcal{E}_\theta^x$**.** In contrast to adjacency matrices, feature matrices are typically much smaller in large networks, making it computationally feasible to extract low-frequency information from node features using standard graph neural networks (GNNs). Previous work (Dwivedi et al., 2023) suggests that the iterative application of the normalized adjacency matrix can serve as a low-pass filter. Inspired by this, we use a Graph Attention Network (GAT) (Veličković et al., 2018) to model the low-frequency components of node features, represented by $\mathcal{X} = \mathcal{E}_\theta^x(G, \mathbf{X})$.

By extracting these low-frequency components from both the graph topology ($\mathcal{E}_\phi^A$) and node features ($\mathcal{E}_\theta^x$), we obtain necessary conditioning information for our diffusion model. Specifically, these components are concatenated to construct the loss function, and Equation (3) is rewritten as:

$$\mathcal{L}_{\text{mse}} = \mathbb{E}_t \left[ \lambda(t)\mathbb{E}_{\mathbf{x}_0}\mathbb{E}_{\mathbf{x}_t|\mathbf{x}_0} \left\| \mathbf{x}_\psi(\mathbf{x}_t, t, C(\mathcal{E}_\phi^A, \mathcal{E}_\theta^x)) - \mathbf{x}_0 \right\|^2 \right] \tag{12}$$

where $\hat{\mathbf{x}}_0 = \mathbf{x}_\psi(\mathbf{x}_t, t, C(\mathcal{E}_\phi^A, \mathcal{E}_\theta^x))$ denotes the reconstructed signal based on the noisy input $\mathbf{x}_t$ and conditioning from the low-frequency encoders. $C(x, y)$ represents the concatenation function, the implementation details are provided in Appendix D.

Although Equation (12) filters high-frequency information early in training, the MSE loss still guides the model toward exact element-wise matching, which reintroduces high-frequency details (Figure 4). This motivates to introduce a new learning objective to address this issue, which will be discussed later in Equations (15) and (16).

## 4.3. Denoise Decoder and the MSS Learning Objective

**Denoise Decoder.** In standard diffusion models, a denoise decoder reconstructs a predicted sample $\hat{\mathbf{x}}_0$ from a noisy version $\mathbf{x}_t$ by minimizing the MSE between $\hat{\mathbf{x}}_0$ and the clean sample $\mathbf{x}_0$. While we also use a denoise decoder for reconstructing clean data, our approach introduces a new objective that treats different frequency components, particularly low-frequency structural signals, with varying priorities.

To implement this, we adopt a U-Net architecture $\mathbf{x}_\psi$ (Li et al., 2024; Dhariwal & Nichol, 2021), where the reconstructed output is given by: $\hat{\mathbf{x}}_0 = \mathbf{x}_\psi(\mathbf{x}_t, t, C(\mathcal{E}_\phi^A, \mathcal{E}_\theta^x))$. As shown in Figure 5(d), the U-Net consists of multiple residual blocks and downsampling layers, followed by a mirrored structure of residual blocks with upsampling layers. We denote the upsampling portion of the network as $\text{NN}^{\text{up}} = \{\text{NN}_1^{\text{up}}, \text{NN}_2^{\text{up}}, \ldots, \text{NN}_l^{\text{up}}\}$. The outputs from these upsampling layers are collected as:

$$\mathbf{H}^{\text{up}} = \{\mathbf{H}_1, \mathbf{H}_2, \ldots, \mathbf{H}_l\}, \tag{13}$$

and concatenated to form the final representation:

$$\mathbf{H} = \text{Concatenation}(\{\mathbf{H}_i\}_{i=1}^l). \tag{14}$$

This multi-scale feature extraction is particularly beneficial for capturing different levels of structural information across the graph (Dhariwal & Nichol, 2021). More details on the U-Net architecture are provided in Appendix D.

**Multi-Scale Smoothing Learning Objective.** Existing diffusion models minimize element-wise discrepancies between the reconstructed $\hat{\mathbf{x}}_0$ and the clean signal $\mathbf{x}_0$, assuming that higher fidelity leads to better representations. However, as shown in Section 3, this MSE-based objective often overemphasizes irrelevant high-frequency details for downstream tasks.

To address this, we introduce the *Multi-Scale Smoothing* (MSS) loss, which focuses on preferentially reconstructing low-frequency, global features. Starting with the MSE-based objective in Equation (12), we define the MSE loss as:

$$\mathcal{R}_{MSE} = \|\hat{\mathbf{x}}_0 - \mathbf{x}_0\|^2 = \left\| \mathbf{x}_\psi(\mathbf{x}_t, t, C(\mathcal{E}_\phi^A, \mathcal{E}_\theta^x)) - \mathbf{x}_0 \right\|^2 .$$

The MSS loss is formulated by focusing on cosine similarity between feature embeddings at different scales, specifically

*Table 1.* Node classification accuracy (%) of various methods. Best and second-best results are bolded and underlined, respectively.

| CATEGORY | METHOD | CORA | CITESEER | PUBMED | OGBN-ARXIV | COMPUTER | PHOTO |
|---|---|---|---|---|---|---|---|
| *Supervised* | GCN | 81.5±0.5 | 70.3±0.6 | 79.0±0.4 | 71.7±3.0 | 86.5±0.5 | 92.4±0.2 |
| | GAT | 83.0±0.7 | 72.5±0.5 | 79.0±0.3 | 72.1±0.1 | 86.9±0.3 | 92.6±0.4 |
| *Random Walk* | NODE2VEC | 74.8 | 52.3 | 80.3 | - | 84.39 | 89.67 |
| | DEEPWALK | 75.7 | 50.5 | 80.5 | - | 85.68 | 89.44 |
| *Self-Supervised* | DGI | 82.3±0.6 | 71.8±0.7 | 76.8±0.6 | 70.3±0.2 | 84.0±0.5 | 91.6±0.2 |
| | MVGRL | 83.5±0.6 | 73.3±0.5 | 80.1±0.7 | 70.3±0.5 | 87.5±0.1 | 91.7±0.1 |
| | BGRL | 82.7±0.6 | 71.1±0.8 | 79.6±0.5 | 71.6±0.1 | 89.7±0.3 | 92.9±0.3 |
| | INFOGCL | 83.5±0.3 | 73.5±0.4 | 79.1±0.2 | 71.2±0.2 | 88.7±0.4 | 93.1±0.1 |
| | CCA-SSG | 84.0±0.4 | 73.1±0.3 | 81.0±0.4 | 71.2±0.2 | 88.7±0.3 | 93.1±0.1 |
| | GPT-GNN | 80.1±1.0 | 68.4±1.6 | 76.3±0.8 | - | - | - |
| | GRAPHMAE | 84.2±0.4 | 73.4±0.4 | 81.1±0.4 | 71.8±0.2 | 88.6±0.2 | 93.6±0.2 |
| | GRAPHTCM | 81.5±0.5 | 72.8±0.6 | 77.2±0.5 | 54.7±0.2 | 84.9±0.3 | 92.1±0.2 |
| | VGAE | 76.3±0.2 | 66.8±0.2 | 75.8±0.4 | 66.4±0.2 | 85.8±0.3 | 91.5±0.2 |
| | SP-GCL | 83.2±0.1 | 71.9±0.4 | 79.2±0.7 | 68.3±0.2 | 89.7±0.2 | 92.5±0.3 |
| | GRAPHACL | 84.2±0.3 | 73.6±0.2 | **82.0±0.2** | 71.7±0.3 | 89.8±0.3 | 93.3±0.2 |
| | DSSL | 83.5±0.4 | 73.2±0.5 | 81.3±0.3 | 69.9±0.4 | 89.2±0.2 | 93.1±0.3 |
| | DDM | 83.1±0.3 | 72.1±0.4 | 79.6±0.9 | 71.3±0.3 | 89.8±0.2 | 93.8±0.2 |
| *Our* | SDMG *(w/ mask)* | **84.3±0.5** | **73.9±0.4** | 80.0±0.5 | **72.1±0.3** | **91.6±0.2** | **94.7±0.2** |
| | SDMG | 83.6±0.6 | 73.2±0.5 | 80.0±0.4 | 70.6±0.2 | 90.4±0.2 | 94.1±0.2 |

over multiple hops in the graph. The MSS objective is given by:

$$\mathcal{R}_{MSS} = S\left(\mathbf{x}_0, \hat{\mathbf{x}}_0\right)^{w_1} \prod_{k=1}^{\text{hop}-1} S\left(\mathbf{h}_0^{(k)}, \hat{\mathbf{h}}_0^{(k)}\right)^{w_k}, \quad (15)$$

where $S(\cdot)$ denotes cosine similarity, and $\mathbf{h}_0^{(k)}$ and $\hat{\mathbf{h}}_0^{(k)}$ represent the feature embeddings of the target node $v_0$ aggregated over its $k$-hop neighborhood. The weight $w_k$ controls the influence of each hop. Specifically, $\mathbf{h}_0^{(k)}$ and $\hat{\mathbf{h}}_0^{(k)}$ are the original and reconstructed feature representations of the target node $v_0$, aggregated with the features from its $k$-hop neighborhood, respectively. This can be formalized as:

$$\mathbf{H}^{(k)} = (\mathbf{A}_{\text{norm}})^k \mathbf{X}, \quad \hat{\mathbf{H}}^{(k)} = (\mathbf{A}_{\text{norm}})^k \hat{\mathbf{X}}.$$

Parameter settings and analysis of $w_k$ and $k$ are given in Appendices C.2 and E. This formulation ensures that node features are progressively aggregated across multiple hops, capturing multi-scale and smooth structural information. Thus, the objective in Equation (12) is rewritten as:

$$\mathcal{L}_{\text{mss}} = \mathbb{E}_t\left[\lambda(t)\mathbb{E}_{\mathbf{x}_0}\mathbb{E}_{\mathbf{x}_t|\mathbf{x}_0}\mathcal{R}_{MSS}\right]. \quad (16)$$

The MSS loss leverages two key ideas to enhance representation learning. *i*) *First*, it uses cosine similarity for a pairwise reconstruction, aligning feature directions instead of exact values. This is particularly effective for tasks like node classification, where the angular similarity between node representations is more important than precise value-by-value matching. Nodes belonging to the same class typically occupy similar directions in feature space, making

cosine similarity a better alignment criterion for downstream tasks. *ii*) *Second*, the MSS loss enforces multi-scale consistency across neighborhood scales. By considering features aggregated over multiple hops, this approach emphasizes low-frequency signals that are stable across larger graph regions and preserve crucial global structural information. A recent work, GraphMAE, introduces a per-node scaled-cosine error (SCE) loss (Hou et al., 2022); however, unlike SCE, MSS aligns representations over multiple hops, thereby explicitly encouraging global low-frequency structure.

Mathematically, we show that this multi-scale smoothing objective preferentially penalizes errors in low-frequency components, forcing the model to reconstruct global structural information more effectively:

**Theorem 4.1.** *Minimizing the loss function in Equation* (15) *based on* $\hat{\mathbf{X}}$ *encourages the model to reconstruct more low-frequency features with a low-pass filter* $g(\hat{\lambda}_i) = (1 - \hat{\lambda}_i)^k$.

**Proof Sketch.** See Appendix A.2 for the complete proof. Intuitively, multi-scale neighborhood consistency emphasizes slowly varying, *global* features (low-frequency components), while allowing more flexibility in reconstructing high-frequency details that contribute less to the aggregated embeddings over larger neighborhood sizes.

Empirically, we find that MSS leads to a stronger representation quality and an improved classification accuracy, as detailed in the experimental Section 5. Together with the low-frequency encoder loss $\mathcal{L}(\mathcal{A})$ in Equation (11) and $\mathcal{L}_{\text{mss}}$ in Equation (16), the final loss is $\mathcal{L}_{loss} = \mathcal{L}(\mathcal{A}) + \mathcal{L}_{mss}$.

*Table 2.* Graph classification accuracy (%) of various methods. Best and second-best results are bolded and underlined, respectively. "–" denotes data that is out of memory or not included in the original paper.

| CATEGORY | METHOD | IMDB-B | IMDB-M | PROTEINS | COLLAB | MUTAG |
|---|---|---|---|---|---|---|
| *Supervised* | GIN | $75.1_{\pm 5.1}$ | $\underline{52.3}_{\pm 2.8}$ | $\mathbf{76.2}_{\pm 2.8}$ | $80.2_{\pm 1.9}$ | $89.4_{\pm 5.6}$ |
| | DiffPool | $72.6_{\pm 3.9}$ | - | $75.1_{\pm 3.5}$ | $78.9_{\pm 2.3}$ | $85.0_{\pm 10.3}$ |
| *Random Walk* | Node2Vec | $50.20_{\pm 0.90}$ | $36.0_{\pm 0.70}$ | $57.49_{\pm 3.57}$ | - | $72.63_{\pm 10.20}$ |
| | Sub2Vec | $55.26_{\pm 1.54}$ | $36.70_{\pm 0.80}$ | $53.03_{\pm 5.55}$ | - | $61.05_{\pm 15.80}$ |
| | graph2vec | $71.10_{\pm 0.54}$ | $50.44_{\pm 0.87}$ | $73.30_{\pm 0.05}$ | - | $83.15_{\pm 9.25}$ |
| *Self-supervised* | InfoGraph | $73.03_{\pm 0.87}$ | $49.69_{\pm 0.53}$ | $74.44_{\pm 0.31}$ | $70.65_{\pm 1.13}$ | $89.01_{\pm 1.13}$ |
| | GraphCL | $71.14_{\pm 0.44}$ | $48.58_{\pm 0.67}$ | $74.39_{\pm 0.45}$ | $71.36_{\pm 1.15}$ | $86.80_{\pm 1.34}$ |
| | JOAO | $70.21_{\pm 3.08}$ | $49.20_{\pm 0.77}$ | $74.55_{\pm 0.41}$ | $69.50_{\pm 0.36}$ | $87.35_{\pm 1.02}$ |
| | GCC | $72.0$ | $49.4$ | - | $78.9$ | - |
| | MVGRL | $74.20_{\pm 0.70}$ | $51.20_{\pm 0.50}$ | - | - | $89.70_{\pm 1.10}$ |
| | GraphMAE | $\underline{75.52}_{\pm 0.66}$ | $51.63_{\pm 0.52}$ | $75.30_{\pm 0.39}$ | $80.32_{\pm 0.46}$ | $88.19_{\pm 1.26}$ |
| | InfoGCL | $75.10_{\pm 0.90}$ | $51.40_{\pm 0.80}$ | - | $80.00_{\pm 1.30}$ | $\underline{91.20}_{\pm 1.30}$ |
| | SimGRACE | $71.30_{\pm 0.77}$ | - | $75.35_{\pm 0.09}$ | $71.72_{\pm 0.82}$ | $89.01_{\pm 1.31}$ |
| | DDM | $74.05_{\pm 0.17}$ | $52.02_{\pm 0.29}$ | $71.61_{\pm 0.56}$ | $\underline{80.70}_{\pm 0.18}$ | $90.15_{\pm 0.46}$ |
| *Our* | SDMG | $\mathbf{76.03}_{\pm 0.53}$ | $\mathbf{52.5}_{\pm 0.42}$ | $73.17_{\pm 0.16}$ | $\mathbf{82.23}_{\pm 0.35}$ | $\mathbf{91.58}_{\pm 0.28}$ |

## 5. Experiments

### 5.1. Experimental Setup

**Datasets.** We evaluate our approach on node-level and graph-level benchmarks. For node classification, we use six datasets: three citation networks (Cora, CiteSeer, PubMed (Sen et al., 2008)), two co-purchase graphs (Photo, Computer (Shchur et al., 2018)), and the large-scale arXiv dataset from the Open Graph Benchmark (Hu et al., 2020a). For graph classification, we use five benchmarks: IMDB-B, IMDB-M, PROTEINS, COLLAB, and MUTAG (Yanardag & Vishwanathan, 2015). For graph classification tasks, node degrees serve as the initial attributes, which are then one-hot encoded for processing. More details are presented in Appendix C.3

**Baselines and Settings.** We compare our method[1] with 11 state-of-the-art unsupervised and supervised methods, using accuracy as the primary metric. Our experiments are conducted on 4 NVIDIA H100 GPUs. Further details on baselines, and experimental protocols are provided in Appendix C, with additional results in Appendix E.

### 5.2. Node Classification

**Masking Strategy for Node Classification.** Inspired by Soft Diffusion (Daras et al., 2022) and Ambient Diffusion (Daras et al., 2024) in computer vision, we introduce a masking process for node classification tasks. Concretely, given a masking rate $p$, we sample each entry of a mask vector $\text{mask} \in \{0,1\}^d$ i.i.d. from a Bernoulli($p$) distribution. Formally, $\text{mask}_i \sim \text{Bernoulli}(p), \quad \forall i = 1, \ldots, d,$

and apply it to the clean node features $\mathbf{x}_0$ via element-wise multiplication:

$$\mathbf{x}_0 \leftarrow \mathbf{x}_0 \odot \text{mask}. \quad (17)$$

Here, some feature entries become zero before being fed into the conditional encoder. Consequently, when the denoising decoder receives $\mathbf{x}_t$, a portion of the features has already been masked out. Importantly, our training objective still requires reconstructing the *entire* node features, including positions that were masked out and never observed by the decoder. This compels the model to learn a more holistic approximation of the data distribution, filling in missing dimensions even when they are not visible during encoding.

**Results.** Table 1 shows the performance of SDMG and its variant SDMG *(w/ mask)*. Specifically, relative to the pioneering diffusion model-based graph SSL approach, DDM, SDMG consistently outperforms it on 5 datasets, highlighting the effectiveness of our framework. In particular, the masked variant (SDMG *w/ mask*) achieves slightly higher accuracy on most benchmarks, likely because random feature masking forces the denoiser to model missing dimensions and thus learn more robust, holistic node representations, while reducing over-fitting to spurious details. Moreover, SDMG demonstrates competitiveness when compared to fully supervised approaches and advanced self-supervised methods, highlighting that our diffusion-based perspective on graph representation learning is effective.

### 5.3. Graph Classification

To further validate the effectiveness of our approach, we conduct extensive experiments on graph classification tasks. As reported in Table 2, SDMG achieves the best performance

---

[1]Our implementation is available at: https://github.com/JYZHU03/SDMG.

on four out of five datasets. This result indicates the potential of our *SDMG* in driving strong representation learning performance. Moreover, similar to the node classification setting observations, SDMG consistently surpasses fully supervised baselines and self-supervised methods based on contrastive learning, generative modeling, or random walks, highlighting our framework's practical utility and broad applicability.

### 5.4. Ablation Study

*Table 3.* Results of ablation study. Classification accuracy (%) for models: without low-frequency encoders and MSS loss (w/o both), without low-frequency encoders (w/o LE), without MSS loss (w/o MSS), and full model.

| Dataset | w/o both | w/o LE | w/o MSS | Full |
|---|---|---|---|---|
| Photo | 83.65 | 87.42 | 93.46 | **94.73** |
| Computer | 76.42 | 85.86 | 89.46 | **91.64** |
| IMDB-B | 74.05 | 74.53 | 75.20 | **76.03** |
| MUTAG | 90.15 | 90.53 | 90.40 | **91.58** |

We perform an ablation study to validate the effectiveness of our low-frequency encoders (*LE*) and the proposed multi-scale smoothing (*MSS*) loss. Table 3 compares: (i) our full model, (ii) *w/o LE* (replacing the low-frequency encoders with MLPs), (iii) *w/o MSS* (replacing MSS with MSE), and (iv) *w/o both* (removing both).

Our findings show that eliminating either LE or MSS degrades the classification accuracy, indicating each module's unique contribution. Low-frequency encoders extract global patterns that enhance downstream performance, and our MSS loss prevents overfitting high-frequency details. Furthermore, dropping both modules significantly reduces accuracy, highlighting their complementary roles in boosting representation quality. Although removing MSS causes a smaller accuracy drop than removing the LF encoders, MSS still closes a substantial portion of the gap between "w/o both" and the full model.

## 6. Conclusion

In this paper, we introduced the Smooth Diffusion Model for Graph Representation Learning (SDMG), which integrates low-frequency component encoders for node features and graph topology, along with a novel Multi-Scale Smoothing (MSS) loss. By prioritizing low-frequency information, SDMG effectively captures the global structure of graphs and reduces an overemphasis on high-frequency noise, resulting in better graph representation learning. Our experimental results demonstrate that SDMG outperforms existing methods, particularly in tasks sensitive to structural information, such as node classification and graph reconstruction.

This study connects to spectral graph theory and advances in diffusion models, with much potential for future exploration of adaptive, frequency-aware objectives in graph diffusion learning.

## Acknowledgements

Chao Gao and Dongpeng Hou acknowledge the National Natural Science Foundation of China (Nos. 62261136549, 62471403), the Technological InnovationTeam of Shaanxi Province (No. 2025RS-CXTD-009), the International cooperation project of Shaanxi Province (No. 2025GH-YBXM-017).

Junyou Zhu and Frank Hellmann gratefully thank the Federal Ministry of Education and Research (03SF0766), the German Federal Ministry for Economic Affairs and Climate Action (03EI1092A), and the German Research Foundation (DFG) (KU 837/39-2).

Langzhou He and Philip S. Yu are supported in part by NSF under grants III-2106758, and POSE-2346158.

All authors gratefully acknowledge the Ministry of Research, Science and Culture (MWFK) of Land Brandenburg for supporting this project by providing resources on the high performance computer system at the Potsdam Institute for Climate Impact Research.

## Impact Statement

Our work introduces an unsupervised framework that leverages diffusion probabilistic models to learn robust, low-frequency representations on graphs. In principle, our approach could reduce reliance on large labeled graph datasets, benefiting domains that lack substantial annotations (e.g., drug discovery, social network analysis). We believe this has a potential positive societal impact, as it may accelerate research and development where quick and accurate graph insights are required.

On the other hand, generative techniques, including diffusion models, pose general risks if used without proper oversight, such as generating misleading or adversarial graph data. However, our method is primarily designed for representation learning rather than standalone graph generation, mitigating immediate risks of malicious misuse. We encourage practitioners to combine our approach with security checks or domain-specific safeguards if it is deployed in sensitive applications (e.g., financial networks, healthcare). Overall, this research aims to advance the field of machine learning on graphs and does not introduce new, specific ethical or societal threats beyond those commonly associated with generative modeling.

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

# A. Proof of Theorems.

## A.1. Proof of Theorem 3.1.

**Theorem.** *Consider an encoding $\mathbf{Z}$ of $\mathbf{X}$ with a bounded capacity $I(\mathbf{X}; \mathbf{Z}) \leq C$, where $I(\cdot; \cdot)$ denotes mutual information. Assume that $\mathbf{X}$ takes values in the unit ball of an Euclidean space. Let $Y$ be a target variable to be predicted from $\mathbf{X}$, and assume $\mathbf{X}$ can be decomposed as $\mathbf{X} = (\mathbf{X}_s, \mathbf{X}_r)$ such that $\mathbf{X}_r$ is relevant to $Y$ and $\mathbf{X}_s$ is superfluous to predicting $Y$ given $\mathbf{X}_r$, that is $I(\mathbf{X}_s; Y|\mathbf{X}_r) = 0$. Then we have:*

*The minimal mean square error (mmse) for predicting $\mathbf{X}$ from $\mathbf{Z}$ is bounded as*

$$mmse(\mathbf{X}|\mathbf{Z}) \geq var(\mathbf{X}) - \frac{1}{2}I(\mathbf{Z}; \mathbf{X}) = var(\mathbf{X}) - \frac{1}{2}(I(\mathbf{Z}; \mathbf{X}_r) + I(\mathbf{Z}; \mathbf{X}_s|\mathbf{X}_r)) \tag{18}$$

*The information that the encoding $\mathbf{Z}$ carries on the variable $Y$ is bounded by*

$$I(\mathbf{Z}; Y) \leq C - I(\mathbf{Z}; \mathbf{X}_s|\mathbf{X}_r) \tag{19}$$

*Therefore, optimizing the encoding $\mathbf{Z}$ in order to minimize $mmse(\mathbf{X}|\mathbf{Z})$ is partially misaligned with optimizing the encoding for $I(\mathbf{Z}; Y)$. The former benefits from an increase in encoded superfluous information, the latter is penalized by it.*

*Proof.* The first equation is Theorem 10 of (Wu & Verdú, 2011) plus the chain rule. For the second, first note that as $\mathbf{Z}$ is an encoding of $\mathbf{X}$, thus $I(\mathbf{Z}; Y|\mathbf{X}) = 0$, and $X_s$ is conditionally independent of $Y$. Then we have by the chain rule that

$$I(Y; \mathbf{Z}, \mathbf{X}_s|\mathbf{X}_r) = I(Y; \mathbf{X}_s|\mathbf{X}_r) + I(Y; \mathbf{Z}|\mathbf{X}_s, \mathbf{X}_r) = 0 . \tag{20}$$

As $Y$ is conditionally independent of $\mathbf{Z}$ and $\mathbf{X}_s$, it is also conditionally independent of $\mathbf{Z}$ and we have $I(Y; \mathbf{Z}|\mathbf{X}_r) = 0$. Thus $\mathbf{Z} \rightarrow \mathbf{X}_r \rightarrow Y$ is a Markov chain, and by the data processing inequality we have:

$$I(\mathbf{Z}; Y) \leq I(\mathbf{Z}; \mathbf{X}_r) = I(\mathbf{Z}; \mathbf{X}_s, \mathbf{X}_r) - I(\mathbf{Z}; \mathbf{X}_s|\mathbf{X}_r) \leq C - I(\mathbf{Z}; \mathbf{X}_s|\mathbf{X}_r) \tag{21}$$

$\square$

## A.2. Proof of Theorem 4.1.

**Theorem.** *Minimizing the loss function in Equation (15) based on $\hat{\mathbf{X}}$ encourages the model to reconstruct more low-frequency features with a low-pass filter $g(\hat{\lambda}_i) = (1 - \hat{\lambda}_i)^k$.*

*Proof.* Since $\hat{\mathbf{L}} = \mathbf{I} - \mathbf{A}_{\text{norm}}$ is symmetric and positive semidefinite, it admits an eigenvalue decomposition:

$$\hat{\mathbf{L}} = \mathbf{U}\mathbf{\Lambda}\mathbf{U}^\top, \tag{22}$$

where $\mathbf{U} \in \mathbb{R}^{N \times N}$ is an orthogonal matrix of eigenvectors ($\mathbf{U}^\top\mathbf{U} = \mathbf{I}$), and $\mathbf{\Lambda} = \text{diag}(\hat{\lambda}_1, \hat{\lambda}_2, \ldots, \hat{\lambda}_N)$ contains the eigenvalues $0 \leq \hat{\lambda}_i \leq 2$.

Therefore, the $k$-step aggregation becomes:

$$\hat{\mathbf{X}} = (\mathbf{A}_{\text{norm}})^k \mathbf{X} = \left(\mathbf{I} - \hat{\mathbf{L}}\right)^k \mathbf{X} \tag{23}$$

$$= \left(\mathbf{I} - \mathbf{U}\mathbf{\Lambda}\mathbf{U}^\top\right)^k \mathbf{X} \tag{24}$$

$$= \left(\mathbf{U}(\mathbf{I} - \mathbf{\Lambda})\mathbf{U}^\top\right)^k \mathbf{X} \tag{25}$$

$$= \mathbf{U}(\mathbf{I} - \mathbf{\Lambda})^k\mathbf{U}^\top \mathbf{X} \tag{26}$$

$$= \mathbf{U}(\mathbf{I} - \mathbf{\Lambda})^k\tilde{\mathbf{X}}. \tag{27}$$

Here, $\tilde{\mathbf{X}} = \mathbf{U}^\top \mathbf{X}$ denote the graph Fourier transform of $\mathbf{X}$. This expression indicates that each frequency component $\tilde{\mathbf{X}}_i$ is scaled by $g(\hat{\lambda}_i) = (1 - \hat{\lambda}_i)^k$. Since $0 \leq \hat{\lambda}_i \leq 2$, the magnitude $|1 - \hat{\lambda}_i| \leq 1$.

For low-frequency components (small $\hat{\lambda}_i$), $(1 - \hat{\lambda}_i)$ is close to 1, so $g(\hat{\lambda}_i)$ remains near 1 even for large $k$, preserving these components. For high-frequency components (large $\hat{\lambda}_i$), $|1 - \hat{\lambda}_i|$ is small, and $g(\hat{\lambda}_i)$ diminishes rapidly as $k$ increases, attenuating these components.

By minimizing a loss function involving $\hat{\mathbf{X}}$, the model is encouraged to reconstruct the features emphasizing the low-frequency components of $\mathbf{X}$, effectively acting as a low-pass filter that suppresses high-frequency noise. $\square$

## B. Related Works

Graphs emerge naturally in a wide range of physical (Zhu et al., 2021), and social phenomena (Zhu et al., 2024). In climate science, Cai et al. use teleconnection graphs to reveal spatial disparities in Northern Hemisphere heat-wave trends (Cai et al., 2024a). For spreading processes on large interaction networks (Hou et al., 2025; 2024a;b), Liu et al. propose a percolation-based algorithm for diffusion-source inference (Liu et al., 2023b), while a subsequent study by the same authors introduces a minimum-observer strategy that enables faster outbreak sensing (Liu et al., 2024b). Against this backdrop, learning expressive graph representations has become a central problem in machine learning (Xiong et al., 2023). Below we review the advances most relevant to our work and explain how our Smooth Diffusion Model for Graphs (SDMG) both builds on and extends these directions.

### B.1. Self-Supervised Graph Learning.

Driven by the goal of removing dependency on labeled data, self-supervised learning (SSL) has rapidly gained traction. Two predominant strategies emerge:

**(i) Contrastive SSL.** Contrastive methods seek to learn invariant representations by generating multiple *views* or *augmentations* of the input and ensuring that embeddings of positive examples (e.g., different augmented versions of the same node or subgraph) are similar, while pushing apart representations of negatives (Gao et al., 2023a). Early works such as **DGI** (Veličković et al., 2019) and **InfoGraph** (Xu et al., 2021) rely on mutual-information-based objectives, often by corrupting node features and graph topology to create challenging negative samples. Subsequent approaches explore diverse augmentation schemes, ranging from random-walk sampling (Hassani & Khasahmadi, 2020; Qiu et al., 2020) to feature masking or shuffling (You et al., 2020), to mitigate semantic damage from aggressive perturbations. Recent methods like **BGRL** (Thakoor et al., 2021) discard explicit negatives to simplify training, while **GraphCL** (You et al., 2020) systematically designs graph-level transformations. (Chen et al., 2024b) proposes effective graph contrastive operations in the discrete hamming space. Despite promising results, contrastive methods can be sensitive to the choice of augmentations and sampling strategies, risking suboptimal performance when augmentations are misaligned with downstream tasks.

**(ii) Generative SSL.** Generative strategies instead learn by masking or corrupting part of the graph data (features or edges) and reconstructing the masked information. **VGAE** (Kipf & Welling, 2016b) adopts a variational autoencoder to recover adjacency signals, while **GraphMAE** (Hou et al., 2022) implements a masked autoencoder to reconstruct node features with a noise-robust objective. Other approaches, such as **GPT-GNN** (Hu et al., 2020b), follow an autoregressive paradigm analogous to language models, iteratively predicting node or edge attributes in large-scale or heterogeneous graphs. Although these generative methods effectively capture structural and semantic properties, many give equal priority to all frequency components of the graph signals, potentially overfitting high-frequency noise. Moreover, they often employ heuristic corruption strategies that may not align perfectly with subsequent learning objectives, creating opportunities for improvement in balancing reconstruction fidelity and downstream performance.

### B.2. Diffusion Models for Graphs.

**Background on Diffusion Models.** Diffusion probabilistic models have shown remarkable success in image and text domains by gradually injecting noise into the data, then learning to reverse this process via a denoising network (Ho et al., 2020; Song et al., 2021; Dhariwal & Nichol, 2021; Rombach et al., 2022). Most such methods rely on continuous noise perturbations in the pixel or latent space, enabling high-quality sample generation and powerful representation learning (Tian et al., 2024; Hudson et al., 2024). Other works, for example (Pan et al., 2023), propose using mask-based strategies to

learn image representations. However, extending these approaches to *graph-structured* data poses unique challenges due to discrete topologies and node attributes that may not align neatly with Gaussian noise models.

**Diffusion Models for Graph Generation.**    Early attempts at graph-focused diffusion primarily address *graph generation* rather than representation learning (Niu et al., 2020; Jo et al., 2022; Luo et al., 2024; Zhou et al., 2024; Liu et al., 2023a; 2024a; Chen et al., 2023; Ye et al., 2022; Kong et al., 2023; Fu et al., 2024). For instance, Niu et al. (2020) introduce score-based methods to create permutation-invariant graphs, while Jo et al. (2022) employ stochastic differential equations (SDEs) to handle continuous adjacency matrices and node attributes. Yet, these continuous approaches can struggle with fundamentally discrete structures. To mitigate this, Haefeli et al. (2022) design a discrete diffusion model specific to unattributed graphs, and Vignac et al. (2022) propose DiGress to progressively add or remove edges and nodes through a discrete noise schedule. Several recent generators also operate in the spectral domain: SPECTRE Martinkus et al. (2022) combines GANs with spectral conditioning to model low-frequency structure, while GGSD Minello et al. (2025) applies a diffusion process to a truncated set of eigenpairs before reconstructing the graph adjacency. Such efforts underscore the versatility of diffusion processes for generating novel graphs (e.g., molecular or social), but they seldom target *downstream classification* or other tasks that require learned representations.

**Transition to Representation Learning.**    While several diffusion models now integrate autoencoder-like architectures (Preechakul et al., 2022; Wang et al., 2023; Zhang et al., 2022; Wei et al., 2023), exploit latent-variable spaces for more expressive generation (Gao et al., 2023b), or explore decoder-only models (Xiang et al., 2023; Chen et al., 2024a), their objectives remain rooted in learning high-quality representations of images. Only recently have researchers begun to explore diffusion models explicitly for *graph representation learning*. One example is **DDM** (Yang et al., 2024), which adapts a denoising framework to node-level encoding and taps intermediate denoising layers for learned embeddings. Despite this conceptual shift, DDM's default mean-squared-error objective can overemphasize high-frequency fluctuations, echoing known pitfalls in graph neural networks (GNNs) (Kipf & Welling, 2016a; Veličković et al., 2018; He et al., 2022) where low-frequency signals are often most relevant to classification. Additionally, concurrent studies have explored the potential of diffusion models for graph self-supervised learning in discrete (Chen et al., 2025) or continuous spaces (Cai et al., 2024b), however, these approaches still face the same challenges as DDM.

In this work, we address the shortcomings of conventional denoising objectives by selectively reconstructing crucial low-frequency information, thereby aligning the diffusion process with the frequency characteristics of real-world graphs. Our method, *SDMG*, bridges the gap between generative diffusion models and task-oriented representation learning, resulting in embeddings that better capture structurally significant and semantically discriminative features for downstream tasks.

### B.3. Frequency Analysis in Graph Learning

A complementary line of work studies the role of graph frequencies in shaping GNN performance (Kipf & Welling, 2016a; Bo et al., 2021). Low-pass filtering has been identified as a key factor in preserving globally meaningful features (Hoang et al., 2021). However, prior attempts often rely on architectural choices (Hoang et al., 2021) (e.g., stacking many convolutional layers) or contrastive learning-based heuristics (Liu et al., 2022) (e.g., adding adjacency-based smoothing) to approximate low-frequency filtering. In contrast, *SDMG* explicitly capitalizes on frequency insights, introducing a novel multi-scale smoothing objective that naturally favors lower-frequency reconstructions. By doing so, our model not only encourages the diffusion process to ignore irrelevant high-frequency components but also offers a more direct and theoretically principled route to extract globally consistent features. To the best of our knowledge, this is the first work to enhance diffusion-model-based graph SSL from a frequency perspective.

## C. Experimental setting

### C.1. Baselines

We compare our proposed *SDMG* against a diverse range of representative graph representation learning methods from three primary categories: *supervised learning*, *random walk-based*, and *self-supervised learning*.

**Supervised Methods.**    These approaches train Graph Neural Networks using labeled data. **GCN** (Kipf & Welling, 2016a) and **GAT** (Veličković et al., 2018) learn node embeddings via spectral filtering and attention mechanisms, respectively. While they achieve competitive results with sufficient annotations, performance may degrade in large-scale or noisier settings

with scarce labels. For graph-level classification, **GIN** (Xu et al., 2018) leverages a sum-based aggregation that is powerful in graph isomorphism tests, whereas **DiffPool** (Ying et al., 2018) introduces a differentiable pooling scheme to hierarchically coarsen graphs. Both methods heavily rely on labeled supervision and are less applicable in unlabeled or weakly labeled scenarios.

**Random Walk-Based Methods.** These methods typically generate node sequences via random walks and then learn embeddings akin to word2vec. Examples include **node2vec** (Grover & Leskovec, 2016), **DeepWalk** (Perozzi et al., 2014), **Sub2Vec** (Adhikari et al., 2018), and **graph2vec** (Narayanan et al., 2017). While they are unsupervised and straightforward to implement, they primarily capture local topological patterns. Higher-order or global dependencies may be overlooked, limiting their effectiveness in more complex or heterogeneous graphs.

**Self-Supervised Methods.** To remove the reliance on manually labeled data, a wide spectrum of graph self-supervised learning approaches has been proposed. We group these methods into three main branches based on their training objectives:

**(i) Contrastive Methods.** By contrasting positive and negative instances, these models encourage consistent embeddings under various augmentations or subviews. **DGI** (Veličković et al., 2019) uses a global–local alignment, **MVGRL** (Hassani & Khasahmadi, 2020) fuses information across multiple graph views, **BGRL** (Thakoor et al., 2021) introduces a bootstrap strategy under an asymmetric encoder–predictor setup, **InfoGCL** (Xu et al., 2021) integrates information-theoretic principles into the contrastive framework, **CCA-SSG** (Zhang et al., 2021) blends canonical correlation analysis with multi-view graph augmentations, **SP-GCL** (Wang et al., 2024) proposes a single-pass solution catering to both homophilic and heterophilic graphs, **GraphACL** (Xiao et al., 2024) adopts an augmentation-free scheme with asymmetric contrast, and **DSSL** (Xiao et al., 2022) refines the contrastive objective by decoupling representation components.

**(ii) Generative Reconstruction.** Another line of work uses partial masking or autoencoding strategies to learn robust graph embeddings. **GraphMAE** (Hou et al., 2022) extends masked autoencoders to graphs by masking node features, **VGAE** (Kipf & Welling, 2016b) leverages a variational autoencoder to reconstruct graph structure, **GraphTCM** (Fang et al., 2024) exploits multi-task learning to capture richer correlations among self-supervised objectives, while **GPT-GNN** (Hu et al., 2020b) and **GCC** (Qiu et al., 2020) follow a large-scale pre-training paradigm, showing adaptability in heterogeneous or complex graph scenarios.

**(iii) Diffusion-Based Modeling.** Although still in its early stage for graph representation learning, diffusion models show promise. **DDM** (Yang et al., 2024) is an initial attempt that leverages a denoising diffusion process in node feature space to reconstruct the original input; however, it largely focuses on a full-spectrum MSE objective. Such an approach can overfit high-frequency details that do not necessarily benefit downstream recognition tasks. Our *SDMG* extends this line of work by explicitly *prioritizing low-frequency information and mitigating irrelevant high-frequency components*, aligning the generative diffusion objective more closely with discriminative goals.

**Summary and Key Differences.** Our *SDMG* inherits the generative perspective yet departs from traditional "reconstruct-everything" paradigms by emphasizing smooth, global (low-frequency) signals that correlate more strongly with downstream tasks. This targeted design combats the common pitfall of overfitting to high-frequency noise, yielding more discriminative representations and consistent accuracy gains across a variety of node- and graph-level tasks.

## C.2. Hyper-paramter Configurations

The hyperparameters employed in our experiments are detailed in Tables 4 and 5. Note that we did not extensively fine-tune these hyperparameters, suggesting that further optimization could potentially enhance the experimental results.

## C.3. Datasets

In this section, we provide a comprehensive overview of the datasets used in our experiments for both *node-level* and *graph-level* tasks. These benchmarks are widely adopted in the graph representation learning community, and they encompass diverse domains and scales to thoroughly evaluate the effectiveness and generality of our proposed *SDMG* framework.

**Node-Level Datasets.** We employ six standard node classification benchmarks: three citation networks (Cora, CiteSeer, PubMed (Sen et al., 2008)), two co-purchase graphs (Amazon Photo, Amazon Computer (Shchur et al., 2018)), and the large-scale arXiv dataset from the Open Graph Benchmark (Hu et al., 2020a).

*Table 4.* Statistics and hyper-parameters for graph classification datasets.

| Dataset | | IMDB-B | IMDB-M | COLLAB | PROTEINS | MUTAG |
|---|---|---|---|---|---|---|
| **Dataset Statistics** | # graphs | 1000 | 1500 | 5000 | 1113 | 188 |
| | # classes | 2 | 3 | 3 | 2 | 2 |
| | Avg. # nodes | 19.8 | 13.0 | 74.5 | 13.0 | 17.9 |
| **Hyper-parameters** | feat_drop | 0.4 | 0.4 | 0.4 | 0.2 | 0.2 |
| | attn_drop | 0.4 | 0.4 | 0.4 | 0.2 | 0.1 |
| | num_head | 2 | 4 | 4 | 8 | 4 |
| | num_hidden | 128 | 512 | 512 | 512 | 512 |
| | num_hop | 5 | 5 | 5 | 5 | 4 |
| | hop_weights | [0, 3, 3, 1, 1, 1] | [1, 1, 1, 1, 1, 1] | [1, 2, 2, 1, 1, 1] | [1, 2, 2, 1, 1, 1] | [1.5, 1, 1, 1, 1] |
| | learning_rate | $1e^{-5}$ | $1e^{-5}$ | $1e^{-5}$ | $3e^{-4}$ | $3e^{-4}$ |
| | norm | LayerNorm | LayerNorm | LayerNorm | LayerNorm | LayerNorm |
| | beta_schedule | Sigmoid | Linear | Const | Linear | Sigmoid |

*Table 5.* Statistics and hyper-parameters for node classification datasets.

| Dataset | | Cora | Citeseer | PubMed | Ogbn-arxiv | Computer | Photo |
|---|---|---|---|---|---|---|---|
| **Dataset Statistics** | # nodes | 2708 | 3327 | 19717 | 169343 | 13752 | 7650 |
| | # edges | 5429 | 4732 | 44338 | 1166243 | 245861 | 119081 |
| | # classes | 7 | 6 | 3 | 40 | 10 | 8 |
| **Hyper-parameters** | num_hidden | 512 | 1024 | 1024 | 512 | 512 | 512 |
| | learning_rate | $4e^{-4}$ | $2e^{-4}$ | $2e^{-4}$ | $2e^{-4}$ | $2e^{-4}$ | $4e^{-4}$ |
| | num_hop | 2 | 2 | 4 | 4 | 5 | 4 |
| | hop_weights | [1, 0.5] | [1, 0.5] | [1, 1, 1, 1] | [1, 1, 1, 1] | [1, 1, 1, 1, 1] | [1, 2, 2, 1] |
| | norm | LayerNorm | LayerNorm | LayerNorm | LayerNorm | BatchNorm | BatchNorm |
| | beta_schedule | Sigmoid | Linear | Const | Linear | Quad | Sigmoid |

- **Cora, CiteSeer, and PubMed** are classical citation networks where each node represents a scientific publication and edges denote citation relationships. Nodes are assigned bag-of-words features derived from document abstracts, and class labels correspond to academic topics.

- **Amazon Photo and Amazon Computer** are co-purchase networks from Amazon, where products (nodes) are linked if they tend to be purchased together. Node features capture product descriptions and categories, with labels reflecting product classes.

- **OGB-arXiv** is a large-scale citation graph from the Open Graph Benchmark. Each node is a paper from arXiv, edges represent citation relationships, and node features are derived from paper abstracts. The labels correspond to the primary subject area of the paper (e.g., cs.LG, stat.ML).

These datasets vary in size, average node degrees, and feature sparsity, enabling a comprehensive assessment of how well our method generalizes across different scales and graph topologies.

**Graph-Level Datasets.** We further evaluate our method on five widely used benchmarks for graph classification: IMDB-B, IMDB-M, PROTEINS, COLLAB, and MUTAG (Yanardag & Vishwanathan, 2015).

- **IMDB-B** and **IMDB-M** are movie collaboration datasets. Each graph corresponds to an ego-network of actors or directors, with a classification label indicating a genre or thematic category.

- **PROTEINS** is a protein interaction dataset, where each graph represents a protein structure. Nodes indicate secondary structure elements, and edges denote spatial or sequential proximity. Labels characterize protein functions or enzyme classes.

- **COLLAB** is derived from collaboration networks in scientific domains. Nodes are authors, edges represent co-authorship, and graph labels reflect the research field or academic community.

- **MUTAG** consists of small molecular graphs, where each node is an atom and edges represent chemical bonds. The binary classification label indicates mutagenic effect on a given organism.

For these graph-level datasets, node degrees are used as the initial attributes and then one-hot encoded for processing, following standard protocols in prior studies (Yanardag & Vishwanathan, 2015). For both the node-level and graph-level benchmarks, we adopt the commonly used public splits to ensure fair comparison with existing baselines (Yang et al., 2024).

## D. Architecture of Denoise U-Net

**U-Net with Graph-Aware Conditioning.** As illustrated in Figure 5(d), our denoising decoder adopts a U-Net-like structure that combines a contracting (down-sampling) path with an expansive (up-sampling) path. Standard U-Net architectures are typically designed for Euclidean domains (e.g., images), which can limit their ability to capture topological information in graphs. To address this, we enhance U-Net by (i) injecting graph-structured *conditional information* (extracted from our low-frequency encoders) and (ii) using GNN or MLP layers inside the U-Net to better preserve graph structural patterns.

**Implementation for Graph Classification.** For graph-level tasks, the contracting path employs GAT layers, which denoise target node features by aggregating representations from neighboring nodes. This strategy leverages attention mechanisms to focus on the most relevant parts of the graph during down-sampling. In the expansive path, the final GNN layers act as the decoder, projecting the denoised node features into latent representation space and further smoothing them among neighbors. Afterward, we feed these latent features into an MLP to produce reconstructed outputs. In essence, the decoder's final representations already integrate the encoder activations through skip connections, allowing a richer synthesis of local and global graph signals.

**Implementation for Node Classification.** For node-level tasks, we replace the GNN layers in both the contracting and expansive pathways with more lightweight MLPs, which reduce computational overhead without compromising performance. Again, we only use the representations from the decoder side, since the skip connections already channel valuable encoder activations.

**Fusion of Low-Frequency Information.** To ensure that low-frequency structural signals are emphasized, we fuse the noisy data $\mathbf{x}_t$ with the low-frequency encodings of topology ($\mathcal{A}$) and node features ($\mathcal{X}$), both derived from our encoders $\mathcal{E}_\phi^A$ and $\mathcal{E}_\theta^x$. Specifically:

- **Graph classification tasks:**
$$\mathbf{h}^{(l+1)} = \mathbf{h}^{(l)} * \mathcal{A} * \mathcal{X}, \tag{28}$$

  where $*$ denotes element-wise multiplication.

  *Rationale.* In graph-level tasks, each node representation needs to capture both local and global context that contributes to the overall graph embedding. Element-wise multiplication, being a stronger gating mechanism, preserves only those features that consistently appear across $\mathbf{h}^{(l)}$, $\mathcal{A}$, and $\mathcal{X}$. This forces the model to highlight global information shared among the node itself and its low-frequency topology/feature conditioning, making it well-suited for graph classification.

- **Node classification tasks:**
$$\mathbf{h}^{(l+1)} = \mathbf{h}^{(l)} + \mathcal{A} + \mathcal{X}. \tag{29}$$

  Here, the signals are combined by additive fusion.

  *Rationale.* For node-level tasks, we are more interested in preserving each node's specific feature contributions, while still injecting complementary low-frequency topology and feature signals. An additive fusion scheme allows $\mathbf{h}^{(l)}$ to retain node-specific details while gradually incorporating the smoothing effects of $\mathcal{A}$ and $\mathcal{X}$ in a less restrictive way compared to element-wise multiplication. This suits node classification, where fine-grained differences at the node level often matter.

These fusion strategies ensure that the model explicitly incorporates the global (low-frequency) graph structure and feature information into each layer in a manner tailored to the prediction task at hand.

In summary, our enhanced U-Net architecture, combined with carefully designed GNN/MLP layers and the masking process, substantially improves the denoising capabilities on graph-structured data. By incorporating low-frequency structural priors

through conditional encoders, our method avoids overfitting high-frequency noise and achieves robust performance in both node-level and graph-level representation learning tasks.

## E. Parameter Analysis: Effect of *hop* Number

In our Multi-Scale Smoothing (MSS) objective (Equation (15)), the parameter $hop$ controls the maximum neighborhood distance over which node representations are compared. Intuitively, each increment in $hop$ extends the receptive field of the similarity terms $S(\mathbf{h}_0^{(k)}, \hat{\mathbf{h}}_0^{(k)})$, thereby encouraging the denoising model to maintain consistency in increasingly larger neighborhoods around the target node $v_0$.

To investigate the influence of $hop$, we conduct node classification experiments on *Computer* and *Photo*, varying $hop$ from 0 to 5. The rest of the hyperparameters remain fixed to isolate the effect of $hop$. When $hop$ =0, the loss reduces to comparing only the target node features $\mathbf{x}_0$ and its denoised counterpart $\hat{\mathbf{x}}_0$. As $hop$ increases, higher-order neighborhood embeddings $\mathbf{h}_0^{(k)}$ come into play, guided by weights $\{w_k\}$ to emphasize or de-emphasize each scale.

**Results and Observations.** Figure 6(a) and (b) illustrate how classification accuracy progresses with increasing $hop$. We observe a **rapid improvement at low hop values**: moving from $hop$=0 to $hop$=1 or 2 yields a substantial accuracy jump, indicating that incorporating even limited neighborhood information helps the model suppress high-frequency noise and maintain local smoothness. As $hop$ grows to 3 or 4, there are **stable gains beyond hop 3**, suggesting that moderate expansions to the receptive field further refine denoised embeddings and reinforce valuable low-frequency components. Finally, we also find **potential diminishing returns** at $hop$=5, where performance gains become marginal, especially on *Photo*, possibly due to increased redundancy or noisy distant neighbors.

**Takeaways.** Increasing $hop$ generally benefits the diffusion-based denoising process by capturing richer global information. However, the optimal setting appears to lie in a sweet spot where neighborhood expansions enhance low-frequency signal capture without overwhelming the model with uninformative high-frequency details from distant regions. Consequently, tuning $hop$ can be an effective strategy for balancing the trade-off between local precision and global context in our MSS objective.

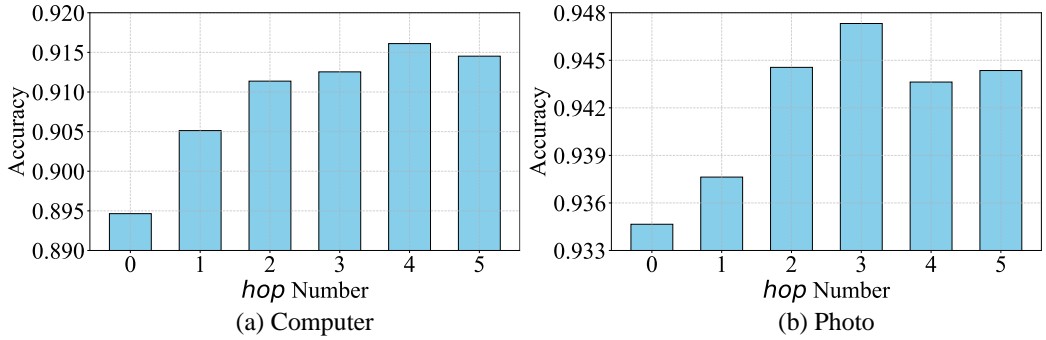

(a) Computer        (b) Photo

*Figure 6.* Classification accuracy versus mask ratio, with each bar presenting accuracy for mask ratios from 0.0 to 0.9.

## F. Limitations and Future Work

While SDMG demonstrates strong performance across multiple benchmarks, our current formulation also reveals a few limitations. First, the framework explicitly prioritizes low-frequency information, potentially underutilizing mid- or high-frequency components that might prove beneficial in certain tasks (e.g., detecting anomalies or fine-grained community boundaries). Thus far, we assume that high-frequency signals are predominantly noise rather than a source of discriminative cues. Extending SDMG to *adaptively* filter or reconstruct different frequency bands—rather than imposing a strict low-frequency bias—may further boost representation quality in domains where subtle high-frequency details matter.

Additionally, although our model avoids computing exact graph spectra by using approximate encoders (thereby mitigating the cubic complexity of full eigendecomposition), some components—such as repeated neighborhood aggregations—could

still become costly on extremely large or dynamically evolving networks. Investigating scalable or incremental strategies for updating low-frequency encoders in streaming scenarios would therefore be a natural direction to handle real-time graph data.

Moving forward, future research could refine the frequency perspective by *selectively* reconstructing mid- or high-frequency signals based on task relevance, or by learning which frequency bands are essential in an *end-to-end* manner. Another intriguing avenue involves unifying our smoothing objective with other forms of structural priors, such as motif- or subgraph-level constraints, to capture richer relational information. Finally, incorporating partial or noisy labels—when they are available—could yield *semi-supervised* extensions of SDMG that seamlessly balance generation and discrimination, ultimately leading to more flexible and robust graph representation learning.

