# OpenReview forum: "SDMG: Smoothing Your Diffusion Models for Powerful Graph Representation Learning"
_ICML.cc/2025/Conference — ICML 2025 poster_

### Official Review · Reviewer_DktW · 2025-03-07

**Overall Recommendation:** 4

**Summary:**

The paper presents a new diffusion-based approach for self-supervised graph representation learning (SDMG). Rather than reconstructing all graph details, which often leads to the overfitting of high-frequency noise, the authors provide both theoretical and empirical evidence that focusing on low-frequency components yields more robust, globally informative representations. To this end, SDMG integrates the proposed low-frequency encoders for node features and graph topology along with a new multi-scale smoothing loss that enforces consistency across multiple neighborhood scales. Extensive experiments on node-level and graph-level benchmarks demonstrate that this targeted reconstruction strategy leads to state-of-the-art performance, while ablation studies validate the contributions of each component, demonstrating the method’s novelty and effectiveness.

**Claims And Evidence:**

The paper’s main claims are supported by solid experimental evidence.

**Essential References Not Discussed:**

The paper provides sufficient related work to understand its key contributions.

I have one minor suggestion: consider discussing a recent vision paper [R1] in ICML 2024 that explores the relationship between recognition and generation. Although that work comes from a different domain and uses a different perspective, it shows that representation and recognition can misalign in visual data. This insight could help readers better understand the misalignment observed in graph data.

[R1] Balestriero, Randall, and Yann LeCun. "How Learning by Reconstruction Produces Uninformative Features For Perception." in ICML 2024.

**Experimental Designs Or Analyses:**

Yes, I reviewed all experimental designs and analyses, particularly the preliminary experiments and ablation studies, and found them to be sound with no significant issues. For example, the conclusions in Figures 1 and 3 are robust and intuitively reasonable since, for recognition tasks, reconstructing every detail is not always beneficial. Recent work in computer vision is just beginning to address this issue from different perspectives, and I am pleased to see such promising prospects in graph tasks. The only minor issue is that the authors should summarize the findings or conclusions more clearly and concisely in the captions of figures and tables.

**Methods And Evaluation Criteria:**

Yes, the methods and evaluation criteria are suited to the problem.

**Other Comments Or Suggestions:**

In page 15, line 773, one citation is not being displayed correctly.

**Other Strengths And Weaknesses:**

**Other Strengths:**

1. I find this work interesting as it highlights a new potential direction for representation learning—designing a suitable loss function to encourage the reconstruction of key aspects rather than all details, which appears promising for improving downstream tasks like classification and prediction.

2. Unlike existing diffusion-based methods, this paper is the first to demonstrate that the graph generation optimization objective is not entirely suited for graph downstream tasks.

3. Instead of using a standard generation-oriented learning objective, the authors propose a new loss function that effectively prioritizes the reconstruction of global and smooth information.

**Other Weaknesses:**

1. In Figure 1, the authors claim that after adding Gaussian noise, the model focusing on a narrow low-frequency band remains robust, while the model with full-spectrum reconstruction collapses in performance. However, the paper does not specify the noise intensity (e.g., the value of σ), which may raise concerns about the robustness.

2. The related work section is placed in the appendix, which might confuse readers who are not familiar with diffusion models or graph learning; this issue could be addressed in the camera-ready version.

3. The current approach focuses only on reconstructing node features, which may limit the model’s ability to capture topological geometry. Exploring the reconstruction of both graph topology and features, for example, by alternating reconstruction [R2], might allow the model to learn more meaningful structural-semantic information。

[R2] Jo, Jaehyeong et.al,. "Score-based generative modeling of graphs via the system of stochastic differential equations." In ICML 2022.

**Questions For Authors:**

1. In Table 1, what does the “–” symbol represent? Does it indicate that the method exceeded available memory or that the computation time was prohibitive?
2. In Equations (11) and (12), did you only concatenate the activations from the U-Net’s up-sampling layers, or did you also include representations from the down-sampling layers? If I understand correctly, the down-sampling information is preserved solely via skip connections into the final representation?
3. In Theorem 3.1, is the encoding “Z” equivalent to the node representation “H” used elsewhere in the paper? If so, it might be clearer to use consistent notation to avoid confusion.

**Relation To Broader Scientific Literature:**

The paper’s key contribution is being the first to reveal the misalignment between graph generation and recognition. Previously, researchers trained generative diffusion models and then extracted the learned representations for classification or prediction. By uncovering this misalignment, the paper could point to a new direction for diffusion model–based graph representation learning.

**Theoretical Claims:**

Yes, I checked the proofs for Theorem 3.1 and Theorem 4.1. They make sense to me: Theorem 3.1 provides strong theoretical support for the observed misalignment between generation and representation learning, while Theorem 4.1 directly validates the equivalence between the proposed MSS loss and a low-frequency filter.

---

> ### Author Rebuttal · Authors · 2025-03-31
>
> We appreciate your valuable input and positive comments, and we have carefully addressed your comments in our responses below.
>
> **Question 1: Clarification on the Meaning of “–” Symbol on Table 1.**
>
> In Table 1, the “–” symbol indicates that the corresponding method either exceeded the available memory, had prohibitive computation time on that particular dataset, or that the original authors did not report performance on that dataset. We will clarify this notation in the revised manuscript to avoid any ambiguity.
>
> **Question 2: Clarification the U-Net Representations on Equations (11) and (12).**
>
> For Equations (11) and (12), we only concatenate the activations from the U-Net’s up-sampling layers. However, the information from the down-sampling layers is fully preserved through skip connections that directly link these layers to the up-sampling path. As a result, although our final representation is built solely from the up-sampling activations, it inherently incorporates the local features captured during down-sampling, ensuring that both local and global information is maintained without redundancy. Thank you for pointing this out, and we’ll revise the final version to make this clearer.
>
> **Question 3: Clarification on the Notation of Encoding “Z” and Node Representation “H”.**
>
> In Theorem 3.1, the encoding “**Z**” typically refers to the intermediate representation produced by our graph encoder. For example, mapping the input graph to a low-dimensional space. On the other hand, “**H**” denotes the final node representation extracted from the activations of the U-Net. In the context of our paper, these two representations can be considered equivalent. Based on your suggestion, in the revised version, we will harmonize our notation by clearly stating that H is derived from **Z**, ensuring consistency throughout the paper.
>
> **Weakness 1: Clarification on Gaussian Noise Intensity Used in Figure 1.**
>
> We appreciate your observation regarding the noise intensity. In our experiments for Figure 1, we sample Gaussian noise from a normal distribution, $\mathcal{N}(0, 5)$, to simulate realistic high-frequency perturbations. We agree that specifying this value explicitly would improve clarity, and we will include these details in the final version. Furthermore, please refer to our response to *Reviewer Q4ro*, Question 1, where we provide a detailed explanation of the motivation behind Figure 1.
>
> **Weakness 3: Clarification on Reconstructing Only Node Features.**
>
> Thank you for your insightful comment. As shown in Figures 3(b) and 3(d), we demonstrate that reconstructing the smooth aspects of the graph structure benefits downstream tasks. Accordingly, in our work we also introduce a low-frequency encoder for graph topology to extract global, smooth structural information. Thus, when reconstructing node features, our SDMG essentially leverages both topology and node feature information to capture the distribution of node features. Nonetheless, we agree that an alternating reconstruction strategy for node features and graph topology is an interesting idea, and we plan to explore this approach in future work.

---

> > ### Comment · Reviewer_DktW · 2025-04-04
> >
> > Thank you for your response and for addressing my concerns.
> >
> > I've re-read the manuscript and found it enjoyable and easy to read. The minor issues I mentioned have been fully addressed and can be easily incorporated into the final version.
> >
> > Compared to Reviewer P6bv's view, I tend to be more open-minded. I understand that low-frequency information or low-frequency filters have proven effective in graph analysis tasks. However, if one brings the existing knowledge to reveal a new phenomenon for a specific domain or a kind of method, especially in the rapidly emerging field of diffusion model-based graph analysis, then that is both interesting and beneficial for community development. To me, a method is not expected to work for every application or dataset type, nor does it have to be overly sophisticated. A good recent example is [1], which uses low-frequency information to inspire researchers to rethink how to build an effective augmented graph for graph contrastive learning.
> >
> > Therefore, I agree with the authors' contributions and believe this paper shows a promising way for future generative or diffusion models for graph analysis. It could motivate people to rethink which parts of the graph information should be reconstructed and which application-specific manifolds should be extracted to better align generation with representation.
> >
> > Given this, I lean toward accepting this paper and will keep attitude to support its acceptance.
> >
> > [1] Liu, Nian, et al. "Revisiting graph contrastive learning from the perspective of graph spectrum." Advances in Neural Information Processing Systems 35 (2022): 2972-2983.

---

> > > ### Author Response · Authors · 2025-04-07
> > >
> > > We greatly appreciate your encouragement and recognition of the novelty of our methodology. We will update our manuscript to enhance its overall clarity. Thank you again for your support!

---

### Official Review · Reviewer_9RUt · 2025-03-07

**Overall Recommendation:** 4

**Summary:**

The authors reveal that purely generation-oriented objectives can conflict with recognition goals, demonstrating that excessive non-smooth-frequency reconstruction can harm representation quality. Specifically, they systematically investigate how reconstructing different parts of the graph frequency spectrum affects downstream classification tasks. Their findings show that reconstructing the full spectrum significantly decreases classification performance. Due to the computational burden of direct spectral decomposition, the authors propose approximate methods, such as a novel low-frequency encoder and a multi-scale smoothing loss, focusing on reconstructing the spectral components most relevant to downstream tasks. Both theoretical analysis and empirical results demonstrate the effectiveness of the proposed approaches.

## update after rebuttal
The authors have adequately addressed my concerns; therefore, I would like to maintain my positive evaluation.

**Claims And Evidence:**

The submission’s claims are validated by evidence from both theoretical analysis and extensive experiments.

**Essential References Not Discussed:**

I did not notice any crucial references missing.

**Experimental Designs Or Analyses:**

The experiments are clear and sound. However, I was a bit confused about the masking strategy experiments presented in the Appendix because I didn’t fully understand their motivation. Please see my questions for more details.

**Methods And Evaluation Criteria:**

The proposed methods make sense for this task. The evaluation criteria rely on standard graph and node classification datasets, which are commonly used benchmarks in representation learning.

**Other Comments Or Suggestions:**

Some figure captions (e.g., Figure 2) have small font sizes that might affect legibility.

**Other Strengths And Weaknesses:**

Strengths 1: It’s the first work to address the key challenge of misalignment between reconstruction and classification in the graph domain.

Strengths 2: The introduction of the multi-scale smoothing loss and low-frequency encoders is innovative and interesting.

Strengths 3: The work demonstrates an interesting finding that existing low-frequency filters in the denoise decoder do not prevent the reconstruction of irrelevant high-frequency features, which motivates the new learning objective.

Weakness 1: A minor weakness is that there is not enough discussion about the motivation for the masking strategy. I detail it more in the "Questions" part.

Weakness 2: As shown in Table 3, the improvement provided by introducing low-frequency encoders appears to be greater than that from the MSS loss. This observation is interesting, but the paper lacks sufficient exploration and explanation of this point, which might raise concerns about the proposed learning objective. Moreover, these observations lead me to wonder whether applying additional low-frequency encoding (such as GNN layers) to the final node representations might be beneficial, or if the design of the MSS loss (Equation 13) already serves the function of GNN layers?

**Questions For Authors:**

Question 1: What is the motivation behind the masking strategy? Specifically, how does the masking interact with the low-frequency reconstruction process, and why is it expected to improve performance on node-level tasks?

Question 2: If I understand correctly, Section 3 refers to the "20% lowest low-frequency components" as the first 20% of the full spectrum. Does this imply that, in Equation (6), $q$=$d$*0.2? If not, please explain how $q$ is calculated.

Question 3: What modifications would be needed to extend the framework beyond classification, such as social community detection or traffic flow prediction?

**Relation To Broader Scientific Literature:**

Comment 1: As far as I know, DDM (Yang et al., 2024) is the first diffusion model-based graph representation learning method. However, it focuses on generation-based reconstruction and may suffer from misalignment with recognition tasks. Unlike DDM, the submission work shifts the focus to low-frequency signals, effectively aligning the diffusion process with downstream objectives.

Comment 2: While low-frequency features have been shown to benefit existing GNN-based classification (Hoang et al., 2021; Liu et al., 2022), this submission work shows that traditional low-pass filters (like GNN layers) do not adequately capture these signals within diffusion models. The paper further addresses this gap by proposing a novel multi-scale smoothing loss, which convinced me that the proposed method is innovative in graph representation learning.

**Theoretical Claims:**

I checked the proofs for both theorems. They are clear and logically sound, and I found no issues.

---

> ### Author Rebuttal · Authors · 2025-03-31
>
> Thank you for your constructive and positive feedback; we appreciate your insights and have addressed your comments below.
>
> **Question 1 and Weakness 1: Clarification on Masking Strategy Motivation.**
>
> The masking strategy has two primary motivations: enhancing robustness and promoting learning discriminative global representations. Recent literature (Daras et al., 2022) on diffusion-based models highlights that carefully designed masking or corruption processes encourage the model to selectively prioritize the most meaningful structural information during reconstruction. Specifically, masking introduces controlled information degradation, forcing the model to reconstruct from limited context, which aligns naturally with our objective of focusing on smooth, globally relevant graph structures. This process prevents the model from simply memorizing trivial local or high-frequency noise features. Consequently, such a masking strategy guides the model to leverage global graph topology, improving performance on downstream tasks through more robust and semantically meaningful representations.
>
> We have also demonstrated the performance of our model without masking; please refer to our response to Reviewer P6bv, Question 1, for detailed comparisons. Based on your suggestion, we will include further clarifications regarding this aspect in the final version of the manuscript.
>
> **Question 2: Clarification on Frequency Component Definition.**
>
> Thank you for the insightful question. Yes, in Section 3 (the Investigation Section), the "5% lowest frequency components" indeed refers specifically to the eigenvalues of the Laplacian matrix derived from the normalized adjacency matrix, which naturally captures the global structure of the graph.
>
> We adopt the Laplacian spectrum because it reflects the smooth, global signals of the graph. Please note that we perform the graph Fourier transform only in the Investigation section to validate our hypothesis. In the Method section, we instead propose an approximate computation method for extracting low-frequency information, which is more efficient and scalable.
>
> **Question 3: How to extend SDMG beyond classification, such as social community detection or traffic flow prediction.**
>
> Extending our framework to tasks such as social community detection or traffic flow prediction would involve a few targeted modifications that build on our core approach of low-frequency emphasis and multi-scale smoothing. For social community detection, one could replace the classification head with a clustering-oriented output by incorporating a clustering loss (e.g., modularity maximization or a spectral clustering regularizer) to directly optimize the learned embeddings for community structure, or by using the generated representations as input to an external clustering algorithm. Additionally, integrating community-aware regularizers into the diffusion process could further enhance the capture of global structural patterns.
>
> For traffic flow prediction, a regression task with significant temporal dynamics, the output layer would need to be restructured to produce continuous predictions (using, for instance, a mean squared error loss), and it would be beneficial to integrate temporal modeling components (such as recurrent layers, temporal convolutional networks, or attention mechanisms) into the denoising decoder. These modifications would allow our method to adapt its emphasis on smooth, global features and multi-scale consistency to effectively address the unique challenges of these different tasks.
>
> **Weakness 2: Clarification on Balancing MSS Loss and Low-Frequency Encoders.**
>
> Thank you for your insightful comment. While Table 3 shows a larger improvement from introducing the low-frequency encoders compared to the MSS loss alone, we believe these components have distinct yet complementary roles. The low-frequency encoders provide a robust initial approximation of the global, low-frequency signals from the graph topology and node features, which are critical for guiding the denoising process. In contrast, the MSS loss fine-tunes the representations by enforcing multi-scale consistency and preventing the reintroduction of high-frequency noise during training. Although its standalone impact may seem smaller, the MSS loss is essential for maintaining overall representation quality over time. Moreover, adding extra low-frequency encoding (e.g., additional GNN layers) to the final node representations would likely be redundant, as our ablation studies show that the best performance is achieved when both the low-frequency encoders and the MSS loss are used together.
>
> Thank you again for your helpful comment. We will incorporate the necessary modifications in the final version.

---

> > ### Comment · Reviewer_9RUt · 2025-04-05
> >
> > The authors have addressed my concerns, and I appreciate the additional clarification on the motivation behind the masking strategy. I have also reviewed the comments from other reviewers and found the analysis compelling—particularly the part demonstrating that even without masking, the model maintains strong performance on both node- and graph-level classification tasks. In my view, even without the masking strategy, the proposed combination of LF and MSS effectively mitigates the misalignment between generation and GRL, and is sufficiently novel and valuable.
> >
> > Overall, I remain positive about this work and maintain the positive score.

---

> > > ### Author Response · Authors · 2025-04-07
> > >
> > > Thank you so much for your efforts during the review phase and for maintaining a positive score. We will reorganize the mask strategy in the final version. Thanks again!

---

### Official Review · Reviewer_Q4ro · 2025-03-10

**Overall Recommendation:** 4

**Summary:**

The authors introduce a new diffusion-based self-supervised framework for graph representation. It addresses the issue that minimizing generation-based learning objectives can overfit high-frequency noise instead of capturing important global structures. To overcome this, the authors propose (1) learnable encoders that approximate low-frequency components of both node features and graph topology, and (2) a new multi-scale smoothing (MSS) loss emphasizing multi-hop similarity for more semantically meaningful information. Experiments show the proposed method outperforms baselines in node-level and graph-level tasks.

**Claims And Evidence:**

The paper’s main claims on the importance of low-frequency components and the effectiveness of SDMG are backed by thorough experiments and theoretical arguments across multiple benchmark datasets.

**Essential References Not Discussed:**

No clearly essential prior works appear to be omitted.

**Experimental Designs Or Analyses:**

The experimental design and analyses mainly follow standard practices (DDM in 2024) and appear valid with no evident methodological flaws.

**Methods And Evaluation Criteria:**

Yes. The authors verify their method on widely accepted node and graph classification benchmarks (Cora, Citeseer, PubMed, etc.).

**Other Comments Or Suggestions:**

None.

**Other Strengths And Weaknesses:**

Strengths:

The paper’s theoretical and experimental analysis effectively demonstrates how pure MSE-based reconstruction can diverge from learning discriminative representations.

Applying diffusion models to graph representation learning is an emerging yet crucial research topic that remains relatively underexplored.

The fusion of a spectral perspective with diffusion modeling introduces a novel angle for graph learning field.

Weaknesses:

The multi-hop neighborhood weights and choice of how many low-frequency components to encode may require careful tuning on different graphs.

Mainly emphasizing low-frequency signals might not be optimal for tasks that rely on high-frequency information (e.g., anomaly detection or graph isomorphism), and although the authors briefly mention the potential value of mid/high-frequency components, the current approach remains mainly focused on low-frequency prioritization.

**Questions For Authors:**

Can you provide more details on the experimental setup in Figure 1? Specifically, why did you add noise in Figure 1(b), and what variance did you use for the Gaussian noise?

In real-world applications such as biomedicine, interpretability can be crucial. Do you think the key frequencies, like the low-frequency components shown to be critical, could offer interpretive insights into the model’s outputs?

In the $R_{MSS}$ loss proposed in Eq. (13), is the construction of $h_0$ the same as in existing GNN layers, or does it employ a new aggregation mechanism?

**Relation To Broader Scientific Literature:**

The paper extends prior diffusion-based generative models into frequency-aware graph representation learning. It bridges concepts from spectral GNNs with recent advances in diffusion-based self-supervision. This integration builds on findings about overfitting to high-frequency noise in graph tasks and proposes a new approach by introducing novel multi-scale smoothing loss with low-frequency encoders. This work provides a novel perspective for existing diffusion model-based graph learning works.

**Theoretical Claims:**

The authors provide two core theorems on how low-frequency reconstruction aligns with improving representations. The proofs are clearly structured and do not show obvious errors; they seem consistent with known results in the field.

---

> ### Author Rebuttal · Authors · 2025-03-31
>
> Thank you for your positive feedback. We appreciate your time and have addressed your comments below.
>
> **Question 1: Clarification on noise addition in Figure 1.**
>
> The primary goal of Figure 1(a) is to illustrate the misalignment between generative reconstruction objectives and representation learning. As shown, reconstructing only a small fraction of information (e.g., the lowest 5% frequency components) is sufficient to achieve competitive downstream performance. Conversely, reconstructing the full frequency spectrum (100%), as most current methods do, results in performance degradation. We attribute this to diffusion models allocating their limited information capacity to unnecessary reconstruction of high-frequency noise.
>
> To further confirm this hypothesis, we performed the experiment shown in Figure 1(b), where we explicitly added high-frequency Gaussian noise with variance $\sigma^2 = 5$, specifically sampling from $\mathcal{N}(0, 5)$. The rationale is that, if the diffusion model could effectively prioritize reconstructing task-relevant (low-frequency) information under limited capacity, its performance should remain robust despite the addition of irrelevant high-frequency noise.
>
> Indeed, in Figure 1(b), we observe that models focusing solely on reconstructing a small fraction of low-frequency information remain robust, maintaining strong downstream performance even after noise injection. In contrast, reconstructing the full frequency spectrum leads to a significant performance drop under added noise conditions. This clearly indicates that under limited capacity, enforcing full reconstruction exacerbates the misalignment between generation and representation. Our theoretical analysis in Theorem 3.1 further supports and explains this phenomenon.
>
> Thank you for your insightful comment. We will revise the final version accordingly.
>
> **Question 2: Clarification on Interpretability implications of focusing on low-frequency signals.**
>
> Low-frequency components summarize global structural information and thus can offer valuable interpretive insights in **certain** applications. For instance, in relatively homogeneous networks such as social networks, low-frequency signals help explain group memberships in clustering tasks by highlighting shared global characteristics among nodes. However, interpretability is inherently task- and context-dependent. In applications like biological anomaly detection or extreme event prediction, relying solely on low-frequency information may not provide sufficient interpretability, as critical insights could reside in higher-frequency or local features.
>
> **Question 3: Clarification on the Aggregation Mechanism in MSS loss.**
>
> The construction of $\mathbf{h}$  in the  $R_{MSS}$ loss (Eq. 13) indeed employs an aggregation strategy similar to those used in standard GNN layers. Note that, we leverage this aggregation within our proposed MSS loss specifically to capture multi-scale reconstruction information across different neighborhood ranges. By explicitly comparing node representations at multiple neighborhood scales, our MSS loss uniquely encourages the model to prioritize smooth, global (low-frequency) signals during reconstruction, which differentiates it from conventional aggregation strategies that primarily aim for node-level feature updates.
>
> **Weakness: Clarification on Frequency selection and potential extensions.**
>
> We acknowledge the reviewer’s point regarding tasks that might benefit from high-frequency information. While our current approach intentionally emphasizes low-frequency signals due to their effectiveness in mitigating the misalignment between generation and representation, we do not exclude the potential value of mid- or high-frequency components. Instead, we view our method as providing a foundational perspective that future studies can adapt or extend based on specific task requirements. Future work may incorporate adaptive frequency selection mechanisms, which could better accommodate scenarios such as anomaly detection or extreme event prediction.

---

> > ### Comment · Reviewer_Q4ro · 2025-04-04
> >
> > Thank you to the authors for their response. Their explanations are clear and address all points well, providing additional analysis and details. The authors offer a detailed explanation of how the multi-scale smoothing loss works in their model, which convinces me of its novelty in mitigating the misalignment issue. I will maintain my positive score and recommendation for acceptance.

---

> > > ### Author Response · Authors · 2025-04-07
> > >
> > > We greatly appreciate your careful review! Thank you so much for giving us a positive score!

---

### Official Review · Reviewer_P6bv · 2025-03-16

**Overall Recommendation:** 2

**Summary:**

The authors study the problem of graph representation learning (GRL) via diffusion models. The authors argue that current graph diffusion models for representation learning are sub-optimal due to their focus on the high-frequency signals. However previous literature, including a preliminary study by the authors, show that low frequency signals are more important for GRL. As such, the authors design a newer diffusion model that emphasizes lower graph frequencies. This is done through the use of two components. The first is a lowe frequency encoder that attempts to to only synthesize the low frequency dignals of the node features and graph structure. The second component is a multi-scale smoothing loss that places more emphasis on reconstructing lower frequency signals. The authors report their performance on node and graph classification tasks, showing good performance.

**Claims And Evidence:**

The claims are mostly well supported by evidence. This includes both the preliminary study (Finding 1 and 2 in Section 3) and both proposed components (see the ablation study in Table 3).

However, I think the authors should note that the assumption that only the low frequency signals are crucial doesn't hold true for all graphs. For example, for heterophilic graphs, it's known that incorporating high frequency signals is helpful [1].

[1] Luan, Sitao, et al. "Revisiting heterophily for graph neural networks." Advances in neural information processing systems 35 (2022): 1362-1375.

**Essential References Not Discussed:**

A number of references are missing.

For graph generative models, multiple models consider the graph spectrum. This includes [1, 2] (note that [2] was available since early 2024). To be clear, these methods are not designed for GRL. However, they are still generative models that operate on the graph spectrum and should be cited. Furthermore, these methods focus on the lowest k eigen-values/vectors. Of particular interest is [2] which is a diffusion model. The paper should be updated with a discussion of these papers.

Furthermore, the use of the $q$ smallest eigenvectors of the laplacian is quite common in graph literature. See [3]. From my understanding SMDG relies on approximation of these eigenvectors, but the motivation and principle remains the same.


[1] Martinkus, Karolis, et al. "Spectre: Spectral conditioning helps to overcome the expressivity limits of one-shot graph generators." International Conference on Machine Learning. PMLR, 2022.
[2] Minello, Giorgia, et al. "Generating Graphs via Spectral Diffusion." The Thirteenth International Conference on Learning Representations, 2025.
[3] Rampášek, Ladislav, et al. "Recipe for a general, powerful, scalable graph transformer." Advances in Neural Information Processing Systems 35 (2022): 14501-14515.

**Experimental Designs Or Analyses:**

The experimental results and analyses are mostly fine.

However, I think there are a few crucial issues:
1. For node classification, the authors use an additional masking strategy to improve performance. This is quite problematic in my opinion, as Figure 6 shows it can have quite a large effect on performance (I assume mask\%=0 corresponds to an ablation of it.) Without it, the performance is very similar to DDM, another diffusion based approach which as far as I can tell doesn't use any masking. To me, this calls into question if most of the improvement of SDMG over DDM on node classification is due to this inclusion.
2. The performance improvement over simpler methods is quite small and often seems not statistically significant. For example, GraphMAE can achieve only slightly worse performance than SDMG despite being much simpler. As such, I recommend including whether each of the results are better than baselines in a statistically significant way (i.e., include the p-values).
3. No efficiency experiments are shown. It is well known that diffusion models are quite slow. As such, it's important for the authors to show the runtime needed for training + inference of SDMG. Because even if a model can achieve better performance, poor efficiency can make them impractical to realistically use. As such, detailing its efficiency against simpler methods like GraphMAE is important.

**Methods And Evaluation Criteria:**

The baselines and evaluation make sense for this task and are commonly used for GRL.

**Other Comments Or Suggestions:**

1. Please include a mention of the node masking in the main text.
2. Please explicitly define $\mathcal{E}_{\phi}^x$.

**Other Strengths And Weaknesses:**

There are a few notable weaknesses with the paper in its current form.

1. I find the emphasis on diffusion models to be a little strange. As noted by the authors, the idea that low frequency information is most helpful for graph learning is very well known in Graph ML. Specifically, many works have drawn the connection between GNN and low pass filters. The contributions in this paper reflect thus reflect an interest in focusing on low frequency information. However, this has nothing to do with diffusion. As such, both the LE and MSS components can be included in other architectures as well and are not specific to diffusion models. For example, I see no reason why they can't be incorporated into a method like GraphMAE (please correct me if I'm wrong). I'm therefore quite confused by the framing of this paper as I'm unsure what it has to do specifically with diffusion models. To me, it seems like it would apply to any generative methods.

2. Building on the previous weakness, I find the novelty to thus be quite low. This is particulary true in regards to the encoders. $\mathcal{E}\_{\theta}^x$ is just a GNN which is used by all frameworks. $\mathcal{E}\_{\phi}^x$ considers an approximation of the $q$ smallest eigenvectors. Many works consider pre-computing the $q$ smallest eigenvectors of the laplacian and using them as input to a GNN/Graph-Transformer [1]. Therefore, these contributions offer very little novelty over existing methods.

3. In my opinion, some crucial information is ommitted from the main text and either not mentioned entirely or put in the appendix. The main text should be self-contained (within reason of course). Some examples include: **(a)** What is the design of $\mathcal{E}\_{\phi}^x$ exactly? In 4.2, the authors say that they use a neural network function $f$? What is the design of $f$, what is it's input? Furthermore, how is $\mathcal{A}$ defined, is it simply a set of learnable embeddings? **(b)** For node classification, the authors use an additional masking strategy to improve performance. However, this is not mentioned in the main text at all. I expound on this more in the "Experimental Designs Or Analyses" section.


[1] Rampášek, Ladislav, et al. "Recipe for a general, powerful, scalable graph transformer." Advances in Neural Information Processing Systems 35 (2022): 14501-14515.

**Questions For Authors:**

1. What is the performance of SDMG on node classification w/o the node masking? This allows for a fairer comparison with DDM.
2. Can you include the p-values of SDMG's improvement over other methods like DDM and GraphMAE?
3. Can the contributions in this paper be applied to other generative frameworks? If not, why is it specific to diffusion models?
4. Can you show a runtime comparison of SDMG compared to other methods (e.g., GraphMAE)?

**Relation To Broader Scientific Literature:**

I think this paper is relevant for the field of graph representation learning. Specifically the emphasis on the low frequency signals is a useful insight.

**Theoretical Claims:**

The theoretical claims in Section 3 look good to me.

---

> ### Author Rebuttal · Authors · 2025-03-31
>
> We thank you for taking the time to review our manuscript and for the suggestions that have helped clarify our original contributions. Due to its overarching importance, we would like to start with Weakness 2:
>
> **Novelty**
>
> Our work is not on GRL as such, but GRL for use in a downstream classification task. As highlighted by Reviewers 9RUT and DKTW we are the first to consider potential misalignment and possible solutions in this context. Specifically：
>
> 1. **Identifying and analyzing the misalignment between “graph generation” and “graph representation”.**
>    - Existing diffusion-based GRL typically seeks to match the entire frequency spectrum, including irrelevant details. From both **theoretical and empirical** perspectives, we show that this reduces performance on downstream tasks.
>
> 2. **Proposing a novel **multi-scale smoothing (MSS) loss** that mitigates misalignment between generation and discriminative representation.**
>    - Even if one applies known LF filters (e.g., by truncating high frequencies or using a GNN), relying on MSE for element-wise reconstruction reintroduces high-frequency noise. (see Figure 4 and Theorem 3.1). Our MSS loss overcomes this by enforcing stronger penalties on LF errors throughout the diffusion steps. This mechanism is new and, from a **theoretical** standpoint (Theorem 4.1), leads the reconstruction toward LF signals without entirely suppressing useful high-frequency components.
>
> Thus, we emphasize that the **key contribution** is **not** about which LF encoder one employs (we use a mature and efficient GNN simply because it is practical). Rather, it is **the combination of** LF and our MSS that resolves the inherent misalignment between generation and representation.
>
> We thank you for your comments and will revise our manuscript to reflect this more clearly.
>
> **Question 1 and Issue 1: Performance Without Masking**
>
> Below are our node/graph classification results without any masking. As shown, our method consistently improves over DDM on all datasets, with especially substantial gains in graph classification.
>
> *Node Classification*
> |Method|Cora|Citeseer|Pubmed|OGB-Arxiv|Computer|Photo|
> |-|-|-|-|-|-|-|
> |DDM|83.1|72.1|79.6|71.3|89.8|93.8|
> |SDMG w/o mask|83.6|73.2|80.0|71.7|90.4|94.1|
>
> *Graph Classification*
> |Method|IMDB-B|IMDB-M|PROTEINS|COLLAB|MUTAG|
> |-|-|-|-|-|-|
> |DDM|74.05|52.02|71.61|80.07|90.15|
> |SDMG w/o mask|76.03|52.05|73.17|82.23|91.58|
>
> **Question 2 & Issue 2: Statistical results (*p*-value)**
>
> In our paper, we report the widely used mean and standard deviation as standard performance statistics. However, we are also happy to include more statistical tests (e.g., Wilcoxon Signed-Rank Test) with p-values.  In general, p < 0.05 indicates statistically significant improvements; while some datasets (e.g., Cora) show p > 0.05, most meet p < 0.05.
>
> |SDMG (Ours) vs.|Computer|Photo|Cora|MUTAG|COLLAB|IMDB-B|
> |-|-|-|-|-|-|-|
> |DDM|0.00098|0.00488|0.01367|0.00195|0.00098|0.00195|
> |GraphMAE|0.00098|0.00098|0.21582|0.00098|0.00098|0.02441|
>
> **Question 3 & Weakness 1: Why Diffusion Model Benefits**
>
> Thanks for your comment. We do not merely “insert” a low-frequency (LF) encoder and a multi-scale smoothing (MSS) module into generative models. Rather, we exploit the diffusion process’s iterative noise-injection–denoising scheme: at each timestep t, varying noise levels enable our MSS term to progressively emphasize coherent LF signals. This contrasts with single-step models (e.g., GraphMAE), which lack such iterative noise scheduling. While our MSS approach can be generalized elsewhere, the table below shows that it yields larger gains in our multi-step diffusion framework, aligning with our core viewpoint.
> ||photo|computer ||photo| computer|
> |-|-|-|-|-|-|
> |GraphMAE|93.6|88.6|SDMG w/o MSS|93.4|89.4|
> |GraphMAE+MSS|93.9|89.4|SDMG|94.7|91.6|
> |Relative Improvement|0.32%|0.90%|Relative Improvement|1.36%|2.44%|
>
> **Question 4 & Issue 3: Runtime Efficiency**
> Our method does not suffer from excessive runtime overhead (training + inference)—in fact, it often converges faster. By prioritizing global smooth signals (e.g., via our MSS), SDMG quickly encodes downstream-relevant information and requires fewer total epochs. All experiments were run on a single NVIDIA H100.
>
> |Method|Cora (epochs)|Citeseer (epochs)|Pubmed (epochs)|
> |-|-|-|-|
> |GraphMAE|13.32s (1500)|3.58s (300)|20.69s (1000)|
> |DDM|13.12s (800)|11.23s (350)|10.35s (300)|
> |SDMG|5.88s (200)|9.39s (150)|15.23s (100)|
>
> **Weakness 3: Notation Clarification**
>
> Thank you for the suggestion. As noted (page 6, line 290), $\mathcal{E}^x_{\theta}$ is a GAT encoder for LF node features.  $f = \mathcal{E}^A_{\phi}$ is an MLP for topology. $\mathcal{A}$ is a learnable embedding matrix approximating the LF spectrum. $f$ can take various graph-structure inputs (e.g., adjacency), and we use random-walk positional encodings for preserving LF information. We will expand these details (including the mask strategy) in the final version.

---

> > ### Comment · Reviewer_P6bv · 2025-04-02
> >
> > I appreciate the detailed response. However most of my concerns still stand. I will therefore keep my score. I respond to each point below:
> >
> > > Identifying and analyzing the misalignment between “graph generation” and “graph representation”
> >
> > My issue with this lies with two points:
> > 1. It's trivial that graph generation attempts to reconstruct the entire frequency spectrum. Since we attempt reconstruct the entire adjacency such signals must, by definition, be what we also attempt to reconstruct.
> > 2. It's already very well known that standard GNNs are low pass filters [1, 2]. In fact, this is why they tend to work so well, as only considering lower frequency signals is a strong inductive bias for many Graph ML tasks. Therefore, it's not surprising that you find that considering considering higher frequency signals hurts performance. It's why (a) standard GNNs work so well, (b) many methods consider the eigenvectors associated with the lowest k eigenvalues as positional encodings.
> >
> > I therefore find the misalignment to be fairly obvious to those in the field and am unsure what new information it is adding.
> >
> > [1] Nt, Hoang, and Takanori Maehara. "Revisiting graph neural networks: All we have is low-pass filters." arXiv preprint arXiv:1905.09550 (2019).
> > [2] Wu, Felix, et al. "Simplifying graph convolutional networks." International conference on machine learning. Pmlr, 2019.
> >
> > > Proposing a novel multi-scale smoothing (MSS) loss that mitigates misalignment
> > > Thus, we emphasize that the key contribution is not about which LF encoder one employs
> >
> > This is fair, however my concern is that the results show that the encoder has a much larger impact on performance than MSS. As shown in Table 3, ablating LF almost always has a larger effect on performance than ablating MSS. The difference is in fact quite large on Photo and Computer. I am therefore uncertain about how signficant of a contribution that the MSS loss really is (I also note later that it is related to the SCE loss in GraphMAE, however this connection is not discussed).
> >
> > > Performance Without Masking
> >
> > Thanks for the results. Looking at the node-level results, it's important to note that the gap between SDMG and DDM drops significantly (I assume the graph-level are the same since there isn't any masking). While the variance (and p-values) aren't included. At a glance, it seems that only Citeseer is statistically significant (please correct me if I'm wrong).
> >
> > This is very important to me, as it shows that when evaluated on equal footing, the results for DDM and SDMG on node classification are barely different.
> >
> > > Issue 2: Statistical results (p-value)
> >
> > My concern still stand for node classification, as the results w/o masking are very similar to DDM (which is needed to have a fair comparison with DDM).
> >
> > > Why Diffusion Model Benefits
> >
> > This doesn't get to the core of my issue. The main motivation of this work that reconstructing higher frequency signals can be detrimental. **However, this observation is true for any generative method that attempts to reconstruct the full adjacency matrix**. As such, it is not unique to diffusion models at all.
> >
> > The authors do mention that they also consider the iterative noise scheduling inherit in diffusion models to measure the MSS loss across noise levels. This is good. However (a) using it for GraphMAE, does help a little (b) it again does not get to my core point which that the motivation behind the MSS loss is not unique to diffusion models.
> >
> > Furthermore, you show the results when applying MSS to GraphMAE. However, GraphMAE, as opposed to many other GRL methods, does not reconstruct the adjacency. Rather it reconstructs the node features. In fact, it includes a similar loss to MSS, which they refer to as "SCE" (see Eq. 2 in their paper). If we remove the terms for hops $k=1$ to $hop-1$ in the MSS loss, they are equivalent. However, the connection is not discussed in your paper. It should be included.
> >
> > > Runtime Efficiency
> >
> > I appreciate the results. However, a core downside of diffusion models is not just their inefficiency, but their inefficiency on **medium-large graphs** (my apologies for not being clearer in my original review). Core/Citeseer/Pubmed are quite small compared to most benchmarks. What's the runtime comparison on larger graphs like ogbn-arxiv? It's necessary to scale any method beyond small graphs.
> >
> >
> > **Other**:
> >
> > A few of my concerns included in my original review were not addressed. I understand the space is limited. I'm mentioning them again as I believe they're important.
> >
> > 1. Multiple methods (see [1, 2] in my review) are generative models that consider the spectrum. [2] is a diffusion model that focuses on the lowest k eigenvectors.
> > 2. Using the eigenvectors associated with the $k$ lowest eigenvals as input to a GNN is common in Graph lit ([3] in my original comment is 1 example). The 1st term in the SDMG loss considers a similar concept (in addition to [2]). This should be noted in the paper.

---

> > > ### Author Response · Authors · 2025-04-07
> > >
> > > We sincerely appreciate your time on this second review and will address your concerns as follows:
> > >
> > > Because this is crucial to our method, we would like to first clarify that our approach **only reconstructs node features rather than the adjacency matrix (see Section 4.1: Reconstruction Objective).**
> > >
> > > We realize that this confusion might be due to deficiencies in our presentation of the Investigation in Section 3, specifically the discussion of reconstructing adjacency derived node features. As this reconstruction is not especially relevant to our overall paper, we are considering removing this task and clarifying the investigation section if the paper is accepted.
> > >
> > > > Q 1: Clarification of Our Main Contribution
> > >
> > > -	We agree that reconstructing the full adjacency matrix covers the entire frequency spectrum, which is why our method reconstructs only the node feature matrix.
> > > -	We also acknowledge that GNNs act as low-pass filters. However, the key difference lies in their objectives: generative models aim to generate, whereas GNNs extract discriminative features. **Although the importance of LF signals is well recognized, our work is the first to systematically analyze this phenomenon within a diffusion generative model framework. We show that using a standard MSE gradually introduces HF noise (Theorem 3.1), *even with GNN filters*, which degrades representation quality.** In contrast, our proposed MSS loss leverages multi-scale similarity to prioritize LF recovery (Theorem 4.1). It bridges the gap between generation objectives and GRL tasks. We believe this mechanism is novel.
> > >
> > > > Q2: Clarification of Ablation performance of LF and MSS
> > >
> > > We thank you for your observation. It's reasonable that the LF encoder has a larger impact on performance. However, this does not conflict with the claim of our contributions. **While the LF encoder captures LF signals, a standard MSE loss in the diffusion process eventually reintroduces noise (Fig. 4 and Theorem 3.1). This is why, even with LF encoders, our MSS loss is still necessary.** Our MSS loss explicitly counteracts this effect by enforcing multi-scale similarity across different neighborhood aggregations, ensuring that the model’s reconstruction remains focused on LF, globally consistent signals.
> > >
> > > > Q3: Clarification between SCE loss and our MSS loss
> > >
> > > Many thanks for highlighting the existing SCE loss. The SCE and our MSS loss are designed for different purposes. The SCE loss, which is designed for sensitivity issues, still operates at a node level and does not incorporate multi-hop or multi-scale neighborhood aggregation, limiting its ability to solve our challenge (i.e., encouraging LF reconstruction). In contrast, our MSS loss leverages multi-scale similarity, which both theory and experiments show encourages LF signal reconstruction. This may explain why our MSS improves GraphMAE GRL performance.
> > >
> > > We will update the discussion about SCE in the final version.
> > >
> > > > Q4: Performance without Masking and statistical results
> > >
> > > Without masking, we acknowledge that node classification gains are modest. However, SDMG still consistently outperforms DDM on all datasets, and the improvement in graph classification is substantial.
> > >
> > > Now, we are happy to supply additional statistical results for SDMG without masking (for fair comparison with DDM).
> > > |SDMG w/o mask vs.|Computer|Photo|Cora|MUTAG|COLLAB|IMDB-B|
> > > |-|-|-|-|-|-|-|
> > > |DDM|0. 00195|0.02441|0.13769|0.00195|0.00098|0.00195|
> > >
> > > > Q5: Applicability to Other Generative Methods
> > >
> > > Many thanks for this comment. We now have more space to clarify.
> > >
> > > **We agree that our proposed strategy could potentially benefit other generative methods, which we believe is also one of its potential advantages.** However, as we mentioned in our previous results, our strategy yields greater improvements within the context of the iterative noise scheduling of diffusion models.
> > >
> > > > Q6: Runtime Efficiency
> > >
> > > As suggested, we report runtime on the larger dataset, showing that SDMG does not incur excessive overhead.
> > > |Method|Arxiv (epochs)|
> > > |-|-|
> > > |GraphMAE|258s (1000)|
> > > |DDM|157s (400)|
> > > |SDMG|204s (300)|
> > >
> > > > Q7: Some References Missing
> > >
> > > We appreciate for highlighting the missing references. We agree that recent generative models operating on the graph spectrum, such as [1] and [2], are relevant for the DM context and will include them in the final version. We will also cite the Graph Transformer method [3] for its use of smallest eigenvectors, although our approach approximates these eigenvectors rather than computing them explicitly. We would also like to emphasize that the core novelty of our work does not lie in the choice of LF encoders (whether via the q smallest eigenvectors or through GNN filters), as these are tools used to address the scientific problem uncovered in our manuscript.
> > >
> > > **We sincerely hope our responses have addressed your concerns. If our clarifications and additional evidence meet your approval, we kindly request that you reconsider your score.**

---

### Decision · Program_Chairs · 2025-05-01

**Decision:**

Accept (poster)

**Comment:**

The paper presents a diffusion based self-supervised method for graph representation learning that can be applied to downstream classification tasks. The main message of the work is that generation-based learning objectives may be misaligned from downstream objectives and can focus too much on high-frequency signal or noise which is not useful. Hence, focusing the representation learning on low-frequency signal better aligns the representation learning with downstream objectives.

Most reviewers appreciated this message and found the paper to be a meaningful contribution, with solid theoretical contributions and good experimental validation. One reviewer noted by the end of discussion that the novelty of the contributions may be limited, since the separation of low and high frequencies is common in graph learning, and that performance increases were overall modest. Other reviewers saw these points more favorably, and in my view are not issues that preclude acceptance. Several reviewers asked for better motivation for the masking strategy, and to clarify how much of the performance improvement is attributed to masking as opposed to other components, which the authors did provide during the discussion.

Given the majority consensus that the contribution is sound and valuable, I am recommending acceptance.